

# IDER: IDempotent Experience Replay for Reliable Continual Learning

**Zhanwang Liu**[1,*]   **Yuting Li**[1,*]   **Haoyuan Gao**[1]   **Yexin Li**[4]

**Linghe Kong**[1]   **Lichao Sun**[3]   **Weiran Huang**[1,2,†]

[1] School of Computer Science, Shanghai Jiao Tong University   [2] Shanghai Innovation Institute
[3] Lehigh University   [4] State Key Laboratory of General Artificial Intelligence, BIGAI

## ABSTRACT

Catastrophic forgetting, the tendency of neural networks to forget previously learned knowledge when learning new tasks, has been a major challenge in continual learning (CL). To tackle this challenge, CL methods have been proposed and shown to reduce forgetting. Furthermore, CL models deployed in mission-critical settings can benefit from uncertainty awareness by calibrating their predictions to reliably assess their confidences. However, existing uncertainty-aware continual learning methods suffer from high computational overhead and incompatibility with mainstream replay methods. To address this, we propose idempotent experience replay (IDER), a novel approach based on the idempotent property where repeated function applications yield the same output. Specifically, we first adapt the training loss to make model idempotent on current data streams. In addition, we introduce an idempotence distillation loss. We feed the output of the current model back into the old checkpoint and then minimize the distance between this reprocessed output and the original output of the current model. This yields a simple and effective new baseline for building reliable continual learners, which can be seamlessly integrated with other CL approaches. Extensive experiments on different CL benchmarks demonstrate that IDER consistently improves prediction reliability while simultaneously boosting accuracy and reducing forgetting. Our results suggest the potential of idempotence as a promising principle for deploying efficient and trustworthy continual learning systems in real-world applications. Our code is available at https://github.com/YutingLi0606/Idempotent-Continual-Learning.

## 1 INTRODUCTION

Deep learning has achieved impressive success across various domains. However, a static batch setting where the training data of all classes can be accessed at the same time is essential for attaining good performance (Le & Yang, 2015; Rebuffi et al., 2017). In many real-world deployments, data arrive sequentially and previously seen samples cannot be fully retained due to storage or privacy constraints. This makes it a major challenge because neural networks tend to rapidly forget previously learned knowledge when trained on new tasks, which is a phenomenon known as catastrophic forgetting (McCloskey & Cohen, 1989).

To address this challenge, continual learning (CL) is proposed to enable models to accumulate knowledge as data streams arrive sequentially. Among valid CL strategies, rehearsal-based approaches are popular as they are simple and efficient. They (Boschini et al., 2022; Buzzega et al., 2020; Caccia et al., 2021; Chaudhry et al., 2019; Wu et al., 2019) address this by storing a small, fixed-capacity buffer of exemplars from previous tasks and replaying them when training on new task, thereby regularizing parameter updates and mitigating catastrophic forgetting. Despite strong average accuracy, CL methods are often poorly calibrated and over-confident, a problem exacerbated by

---

*Equal contribution. This work was conducted at MIFA Lab (members from SJTU & SII).
†Correspondence to Weiran Huang (weiran.huang@outlook.com)

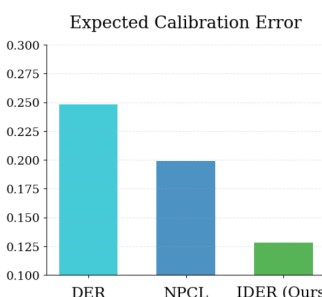 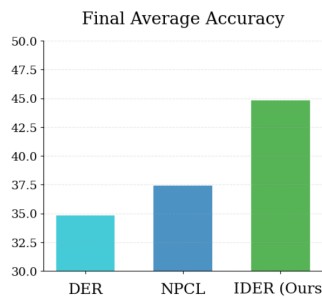 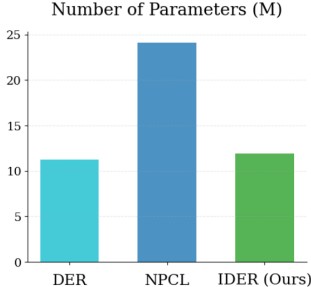

Figure 1: We propose the IDER method, which can be directly applied to many recent rehearsal-based continual learning methods, resulting in less calibration error and significant improvements in FAA with less parameter growth compared with NPCL.

recency bias toward new tasks (Arani et al., 2022). Thus, this undermines the broader deployment of CL models in real-world settings (Li et al., 2024b), especially in safety-critical domains (healthcare, transport, etc.) (LeCun, 2022). CL models deployed in these domains can benefit from uncertainty awareness by calibrating their predictions to reliably assess their confidences (Jha et al., 2024). To tackle this issue, Jha et al. (2023) propose neural processes based CL method (NPCL). However, it causes non-negligible parameter growth and exhibits incompatibility with logits-based replay methods due to the stochasticity in the posterior induced by Monte Carlo sampling. Motivated by these limitations, we aim for a lightweight and compatible principle for reliable CL methods.

We draw inspiration from idempotence, a mathematical property that arises in algebra. An operator is idempotent if applying it multiple times yields the same result as applying it once, formally expressed as $f(f(x)) = f(x)$. It can be used in deep learning by recursively feeding the model's predictions back as inputs, allowing the model to refine its outputs (Durasov et al., 2024a; Shocher et al., 2023). Durasov et al. (2024b) empirically demonstrate that if a deep network $f$ takes as input a vector $x$ and a second auxiliary variable that can either be the ground truth label $y$ corresponding to $x$ or a neutral uninformative signal $0$ and is trained so that $f(x, 0) = f(x, y) = y$, then the distance $||f(x, f(x, 0)) - f(x, 0)||$ correlates strongly with the prediction error. What if we actively minimize this distance of buffer data when we learn new tasks in CL settings? Could we project outputs into the stable manifold where instances are mapped to themselves to prevent predictive distribution drift?

Thus, we propose an Idempotent Experience Replay (IDER) inspired by Idempotence, a simple and effective method that enforces idempotence for CL models when learning new tasks. We demonstrate that enforcing idempotence enables model to make more reliable predictions while reducing catastrophic forgetting. Both combined with naive rehearsal-based method experience replay (ER) (Riemer et al., 2019), compared with NPCL, our approach achieves lower calibration error evaluated by Expected Calibration Error (ECE) (Guo et al., 2017), higher accuracy, and requires smaller parameter numbers, as is shown in Figure 1.

More specifically, IDER integrates two components to enforce idempotence for CL models. Firstly, we adapt the training loss to train the current model to be idempotent with data from the current task. Secondly, we introduce idempotence distillation loss for both buffer data and the current data stream to enforce idempotence between last task model checkpoint $f_{t-1}$ and current model $f_t$. We verify that incorporating the current data stream into idempotence achieves further performance improvements, suggesting that idempotence can help preserve model distribution, thereby mitigating decision boundary drift over time.

This yields a simple method that only requires two forward passes of the model almost without additional parameters. Our approach can be integrated into existing CL methods and experiments show that this simple change boosts both prediction reliability and final accuracy by a large margin. Especially on the Split-CIFAR10 dataset, enforcing idempotence improves the baseline method ER (Riemer et al., 2019) by up to $26\%$, achieving state-of-the-art class incremental learning accuracy. Through extensive empirical validation on challenging generalized class-incremental learning (Mi et al., 2020; Sarfraz et al., 2025), we demonstrate that this simple and powerful principle improves the reliability of predictions while mitigating catastrophic forgetting in real-world scenarios.

The contributions of this paper can be summarized as follows:

- We propose a novel framework for continual learning based on the idempotent property, which is a simple and robust method. Our method demonstrates that fundamental mathematical properties can be effectively utilized to address catastrophic forgetting for CL.
- We show that IDER can be easily integrated into other state-of-the-art methods, leading to more reliable predictions with comparable performance.
- Extensive experiments on several benchmarks demonstrate that our approach achieves strong performance in both mitigating catastrophic forgetting and making reliable predictions.

## 2 RELATED WORK

**Continual Learning.** The goal of continual learning (CL) is to achieve the balance between learning plasticity and memory stability (Wang et al., 2024). Approaches in CL can be divided into three main categories. Regularization-based methods primarily rely on regularization loss to penalize changes in parameter space of the model (Farajtabar et al., 2020; Kirkpatrick et al., 2017). Rehearsal-based Method (Chaudhry et al., 2019) use a memory buffer to store task data and replay them during new task training. Architecture-based methods (Rusu et al., 2016; Wang et al., 2022) incrementally expand the network to allocate distinct parameters for preserving each task's knowledge. Recently, CL has also been studied in more practical settings such as continual post-training of multi-modal LLMs (Wei et al.; Xu et al., 2025; Yan et al., 2026), continual offline RL (Hu et al., 2025a;b), and federated continual learning (Yao et al., 2024). Among them, Rehearsal-based methods are general in various CL scenarios and can be naturally combined with knowledge Distillation (KD) techniques.

The baseline Experience Replay (ER) (Riemer et al., 2019) mixes the current task data with stored samples from past tasks in the memory buffer during training. DER (Buzzega et al., 2020) store old training samples together with their logits and preserve the old knowledge by matching the saved logits with logits obtained by current model. Its improved version XDER (Boschini et al., 2022) improves performance at the sacrifice of computational costs due to sophisticated mechanisms. CLSER (Arani et al., 2022) introduce a fast module for plastic knowledge and a slow learning module for stable knowledge. BFP (Gu et al., 2023) uses a learnable linear layer to perform knowledge distillation in the feature space. SCoMMER (Sarfraz et al., 2023) and SARL (Sarfraz et al., 2025) enforces sparse coding for efficient representation learning. Neural Processes for Continual Learning (NPCL) (Jha et al., 2023) explore uncertainty-aware CL models using neural processes (NPs).

**Idempotence in Deep Learning.** Idempotence is a property of a function whereby the result of applying the function once is the same as applying it multiple times in sequence. Recent work has explored the application of idempotence in deep learning. In particular, it is defined that the results obtained by the model will not change when applying the model multiple times in practice ($f(f(x)) = f(x)$). The Idempotent Generative Network (IGN) (Shocher et al., 2023) firstly proposes this idea in deep learning for generative modeling and it has the capability of producing robust outputs in a single step. Another work ZigZag (Durasov et al., 2024a) introduces idempotence in neural networks for the measuring uncertainty, which is based on IterNet (Durasov et al., 2024b). IterNet proves that for iterative architectures, which use their own output as input, the convergence rate of their successive outputs is highly correlated with the accuracy of the value to which they converge. ZigZag recursively feeds predictions back as inputs, measuring the distance between successive results. A small distance indicates high confidence, while a large one signals uncertainty or out-of-distribution (OOD) data. Recent work ITTT (Durasov et al., 2024c) combines idempotence with Test-Time Training. These works proves the potential of idempotence in deep learning while these works are based on static batch learning.

## 3 METHOD

In this section, we deliver details of the proposed IDER. We first define both class-incremental learning and generalized class-incremental learning settings. Then, we elaborate on how to introduce idempotence in continual learning. Finally, we introduce the overall objective. An overview of IDER is depicted in Figure 5.

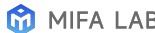

### 3.1 PROBLEM DEFINITION

Continual learning (CL) aims to develop models that learn from a stream of tasks while preserving previously acquired knowledge. In this paper, we focus on typical class-incremental learning (CIL) and generalized class-incremental learning. Generalized class-incremental learning (GCIL) is more close to real-world incremental learning. The key GCIL properties can be summarized as follows: (i) the number of classes across different tasks is not fixed; (ii) classes shown in prior tasks could reappear in later tasks; (iii) training samples are imbalanced across different classes in each task.

In a typical class-incremental learning setting, a model $f$ is trained on sequential tasks $T = \mathcal{T}_1, \mathcal{T}_2, ..., \mathcal{T}_t$ Each task $\mathcal{T}$ consists of data points and these data points are unique within each task, which means $\mathcal{T}_t = \{(x_i, y_i)\}_{i=1}^{N_t}$ and $\mathcal{T}_i \cap \mathcal{T}_j = \emptyset$. The optimization objective is to minimize the overall loss over all the tasks:

$$f^* = \arg\min_f \sum_{i=0}^{t} \mathbb{E}_{(x,y)\sim\mathcal{T}_t} \left[ \mathcal{L}(f(x), y) \right], \tag{1}$$

where $L$ is the loss function for the tasks and $y$ is the ground truth for $x$. However, in the continual setting, only the data from current task $\mathcal{T}_t$ are available and the model should preserve the previous knowledge from the tasks before $\mathcal{T}_1, ..., \mathcal{T}_{t-1}$. As a result, additional memory buffer or additional regularization term $\mathcal{L}_R$ may be chosen to avoid catastrophic forgetting and the actual objective on the current task should be:

$$f^* = \arg\min_f [\mathbb{E}_{(x,y)\sim\mathcal{T}_t\cup\mathcal{M}} \left[ \mathcal{L}(f(x), y) \right] + \mathcal{L}_R]. \tag{2}$$

where $\mathcal{M}$ stands for the memory buffer to store the data from previous tasks.

### 3.2 MODIFIED ARCHITECTURE

To enable idempotence with respect to the second input, we modify the original backbone as shown in Figure 2. We divide the backbone ResNet (He et al., 2016) as denoted $f_t$, into two parts $f_t^1$ and $f_t^2$ on the t-th task. The second input (either a one-hot vector $y$ or a uniform distribution over all classes standing for "empty" input $\mathbf{0}$) is first transformed into a label feature vector. This is achieved by a linear layer with an output dimension that matches the dimensions of $f_t^1$'s output, followed by a LeakyReLU activation function. The image first is processed by $f_t^1$ to produce an intermediate feature map. The label feature is then added to this intermediate feature map, which is fed into $f_t^2$. The output of $f_t^2$, which is the logits for target classes, can work as the second input for model after softmax normalization. In this way, the backbone can accept two inputs and achieve idempotence with respect to the second argument after training.

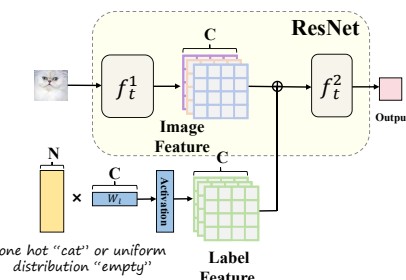

Figure 2: Modified Architecture. We modify the architecture of backbone(ResNet) and enable the model to accept two inputs.

### 3.3 STANDARD IDEMPOTENT MODULE: TRAINING THE NETWORK IDEMPOTENT

First, we rely on the model we train being idempotent. To achieve this, Standard Idempotent Module is used for training the model on data from the current task. Following Durasov et al. (2024a;b), when learning new tasks, we minimize the loss which consists of two cross-entropy losses obtained by the logits from the first and second forward propagation of model and the ground truth $y$ :

$$\mathcal{L}_{ice} = \sum_{(x,y)\in\mathcal{T}_t} [\mathcal{L}_{ce}(f_t(x, y^*), y) + \mathcal{L}_{ce}(f_t(x, f_t(x, y^*)), y)], \tag{3}$$

where $\mathcal{T}_t$ is the current task and $y^*$ is the second input that is set to either the ground-truth one-hot vector $y$ or the neutral "empty" signal input $\mathbf{0}$. We randomly select $y$ with probability $1 - P$ and the neutral "empty" signal input $\mathbf{0}$ with probability $P$. The empty signal $\mathbf{0}$ is defined as a uniform distribution over all classes.

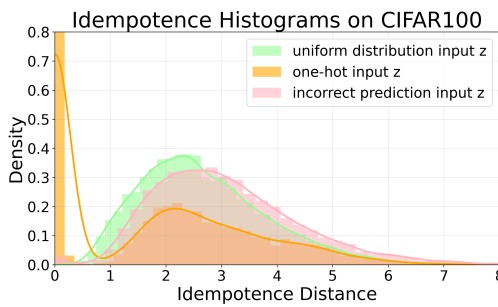

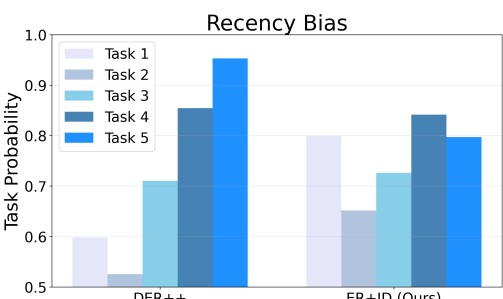

Figure 3: We plot the distribution of idempotence errors, measured by the distance $|f(x, f(x, z)) - f(x, z)|$. Inputs $x$ with second incorrect prediction input $z$ exhibit significantly larger idempotence errors. Thus, this distance can be used as a idempotent distillation loss.

Figure 4: Probability of predicting each task at the end of training for models trained on CIFAR-10 with 500 buffer size. Idempotent distillation loss effectively mitigates the bias to the recent tasks and provides a more uniform probability size across different tasks.

By minimizing $\mathcal{L}_{ice}$, we can train the model idempotent with respect to the second argument, which can be obtained by:

$$f_t(x, \mathbf{0}) \approx y, \quad f_t(x, y) \approx y, \quad f_t(x, f_t(x, \mathbf{0})) \approx y \implies f_t(x, f_t(x, \mathbf{0})) \approx f_t(x, \mathbf{0}). \tag{4}$$

Thus, $f_t$ has been adjusted so that the model $f_t$ is as idempotent as possible for all $x$ in distribution. The model will map the data $(x, \mathbf{0})$ to the stable manifold $(x, y) : f(x, y = y)$. Fig. 3 illustrates this in the case of a network trained on data from the first task on CIFAR-100. With different second input $y$, the idempotence distance distribution varies. The input which contains incorrect prediction input $y$ exhibits significantly larger idempotence errors. Thus, this distance can be used as a distillation loss for iterative prediction refinement to make reliable predictions.

### 3.4 IDEMPOTENT DISTILLATION MODULE: DISTILLING THE NETWORK FOR CONTINUAL LEARNING

In the CL setting, the model tends to have recency bias toward newly introduced classes, which negatively influences the performance and results in overconfidence predictions. Rehearsal-based methods suffer from this problem, as Wang et al. (2022) point out that when a new task is presented to the net, an asymmetry arises between the contributions of replay data and current examples to the weights updates: the gradients of new examples outweigh. To mitigate this issue, we require the model to maintain stable predictions on data from previous tasks even after parameter updates induced by new knowledge, as self-consistency indicates that the network's output is aligned with the learned in-distribution manifold and can make reliable (well-calibrated) prediction. This condition can be translated into enforcing idempotence. Thus, we propose to minimize idempotence distances to mitigate recency bias and prediction distribution drift in CL. A naive way would be to minimize the loss function:

$$\mathcal{L}_{ide} = \sum_{(x,y) \in \mathcal{T}_t, M} \|f_t(x, \mathbf{0}) - f_t(x, f_t(x, \mathbf{0}))\|_2^2. \tag{5}$$

However, this can produce undesirable side effects in CL settings. As $f_t$ has bias towards current data streams and $y_0 = f_t(x, \mathbf{0})$ may be an incorrect prediction, minimizing $\|y0 - y1\|_2^2$ may cause $y_1 = f_t(x, y_0)$ to be pulled towards the incorrect $y_0$, thereby magnifying the error.

To address this, we keep the model checkpoint at the end of the last task $f_{t-1}$ together with the current trained model $f_t$. We then modify the idempotence distillation loss to be:

$$\mathcal{L}_{ide} = \sum_{(x,y) \in \mathcal{T}_t, M} \|f_t(x, 0) - f_{t-1}(x, f_t(x, 0))\|_2^2. \tag{6}$$

Thus, the first prediction $y_0 = f_t(x, \mathbf{0})$ is computed as before, but the second one, $y_1 = f_{t-1}(x, y_0)$, is made using the last model checkpoint $f_{t-1}$ By updating only $f_t$ and keeping $f_{t-1}$ frozen, which preserves more previous knowledge and stable prediction distribution for buffer data, we ensure that $y_0$ is adjusted to minimize the discrepancy with $y_1$, without pulling $y_1$ towards an incorrect $y_0$. Thus,

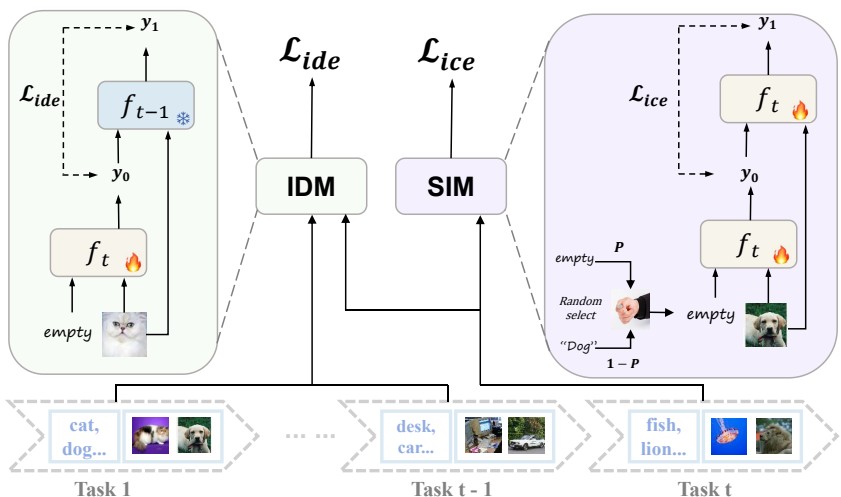

Figure 5: Overall framework of Idempotent Experience Replay (IDER). Our method consists of two modules for continual learning: (1) Standard Idempotent Module that trains current model idempotent with data from the current task. (2) Idempotent Distillation Module that enforce the current model to become idempotent with respect to the last task model checkpoint, utilizing data from both the current task and buffer memory. IDER can be integrated into existing CL approaches to make reliable predictions while mitigate catastrophic forgetting.

we could prevent manifold expansion that would include mistaken outputs, focusing optimization solely on the first pass prediction and thereby avoiding error reinforcement.

This design achieves idempotence by ensuring that processing an input through the current model and then through the last model checkpoint yields a nearly identical output distribution. This self-consistency between the current model and the last model satisfy the requirement for idmpotence condition for sequential tasks in CL. What's more, $\mathcal{L}_{ide}$ can also serve as the distillation loss to mitigate catastrophic forgetting. Unlike traditional distillation in Buzzega et al. (2020), which only aligns the final output probabilities, our method anchors the model's representation to the stable manifold already learned by the frozen model, thereby maintaining balanced predictive performance across all tasks, as is shown in Figure 4. As a result, enforcing idempotence can help CL models mitigate catastrophic forgetting while make reliable preidctions.

## 3.5 OVERALL OBJECTIVE

We introduce idempotence into an experience replay (ER) framework (Riemer et al., 2019), where we keep a buffer $M$ storing training examples from old tasks. We keep the model checkpoint at the end of the last task $f_{t-1}$ together with the current trained model $f_t$. During continual learning, the current model $f_t$ is trained on the batch from data stream of the current task $\mathcal{T}_t$ using the adapted training loss $\mathcal{L}_{ice}$ in Eq. 3. We sample batch from $M$ and combine the current batch to compute idempotence distillation loss $\mathcal{L}_{ide}$ in Eq. 6.

Meanwhile, we sample another batch from $M$ for experience replay. The experience replay loss $\mathcal{L}_{rep\text{-}ice}$ in ER is:

$$\mathcal{L}_{rep\text{-}ice} = \sum_{(x,y)\in M} [\mathcal{L}_{ce}(f_t(x, y^*), y) + \mathcal{L}_{ce}(f_t(x, f_t(x, y^*)), y)]. \tag{7}$$

The total loss function used in IDER is the weighted sum of the losses above, formally:

$$\mathcal{L}_{IDER} = \mathcal{L}_{ice} + \alpha\mathcal{L}_{ide} + \beta\mathcal{L}_{rep\text{-}ice}. \tag{8}$$

In addition, our method is simple and robust, which can be combined with other methods, such as BFP (Gu et al., 2023), to achieve higher performances. Details are shown in the appendix.

Table 1: Comparison of Final Average Accuracy (FAA) across different continual learning methods. All experiments are repeated 5 times with different seeds. Results for SARL (Sarfraz et al., 2025) are from our implementation. The best results are highlighted in blue.The second best results are highlighted in green.

| Method | CIFAR-10 | | CIFAR-100 | | Tiny-ImageNet | |
|---|---|---|---|---|---|---|
| | Buffer 200 | Buffer 500 | Buffer 500 | Buffer 2000 | Buffer 500 | Buffer 4000 |
| Joint (upper bound) | 91.93±0.29 | | 71.15±0.51 | | 59.52±0.33 | |
| iCaRL (Rebuffi et al., 2017) | 58.37±3.51 | 62.49±5.42 | 46.81±0.41 | 52.51±0.44 | 22.53±0.62 | 26.38±0.23 |
| ER (Riemer et al., 2019) | 44.46±2.87 | 58.84±3.85 | 23.41±1.15 | 40.47±0.95 | 10.13±0.39 | 25.12±0.56 |
| BiC (Wu et al., 2019) | 52.61±5.37 | 71.95±1.82 | 37.82±1.67 | 47.17±1.17 | 15.36±1.31 | 18.67±0.57 |
| LUCIR (Hou et al., 2019) | 49.18±7.61 | 65.26±2.54 | 37.91±1.18 | 50.42±0.76 | 28.79±0.51 | 31.64±0.51 |
| DER (Buzzega et al., 2020) | 57.92±1.91 | 68.65±1.82 | 34.83±2.09 | 50.12±0.75 | 15.14±1.29 | 20.35±0.35 |
| DER++ (Buzzega et al., 2020) | 62.19±1.94 | 70.10±1.65 | 37.69±0.97 | 51.82±1.04 | 19.43± 1.63 | 36.89± 1.16 |
| ER-ACE (Caccia et al., 2021) | 62.19±1.67 | 71.15±1.08 | 37.81±0.54 | 49.77±0.34 | 20.42±0.39 | 37.76±0.53 |
| XDER (Boschini et al., 2022) | 64.10±1.08 | 67.42±2.16 | 48.14±0.34 | 57.57±0.84 | 29.12±0.47 | 46.12±0.46 |
| CLS-ER (Arani et al., 2022) | 64.56±2.63 | 74.27±0.81 | 43.92±0.62 | 54.84±1.30 | 30.91±0.59 | 45.17±0.89 |
| SCoMMER (Sarfraz et al., 2023) | 66.95±1.52 | 73.64±0.43 | 39.05±0.79 | 49.42±0.85 | 21.47±0.54 | 37.2±0.70 |
| BFP (Gu et al., 2023) | 68.64±2.23 | 73.51±1.54 | 46.70±1.45 | 57.39±0.75 | 28.71±0.55 | 43.17±1.89 |
| SARL (Sarfraz et al., 2025) | 68.87±1.37 | 73.98±0.46 | 46.69±0.79 | 57.06±0.48 | 28.44±2.30 | 38.83±0.81 |
| **ER+ID(Ours)** | 71.02±1.98 | 74.74±0.42 | 44.82±0.85 | 56.59±0.35 | 29.88±1.15 | 43.05±1.40 |
| **BFP+ID (Ours)** | 71.99±0.98 | 76.65±0.63 | 48.53±0.95 | 57.74±0.64 | 30.62±0.47 | 43.51±0.59 |
| **CLS-ER+ID (Ours)** | 70.32±1.12 | 75.48±0.91 | 47.44±2.0 | 56.36±0.78 | 31.62±0.57 | 46.17±0.22 |

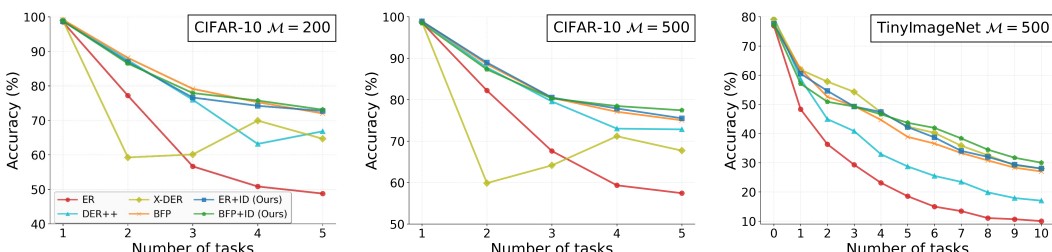

Figure 6: Results on CIFAR-10 and Tiny-ImageNet with different buffer size. It shows the trend of the average test-set accuracy on the observed tasks.

## 4 EXPERIMENTS

**Continual Learning Settings.** We follow Gu et al. (2023) and conduct experiments on state-of-the-art rehearsal-based models in CIL setting which splits the dataset into a sequence of tasks, each containing a disjoint set of classes, while task identifiers are not available during testing. Following Sarfraz et al. (2025), we also evaluate methods in the generalized class incremental learning (GCIL) setting (Mi et al., 2020) which is closest to the real-world scenario as the number of classes in each task is not fixed, the classes can overlap and the sample size for each class can vary.

**Evaluation Metrics.** Following Boschini et al. (2022); Buzzega et al. (2020), we use Final Average Accuracy (FAA) and Final Forgetting (FF) to reflect the performances of mitigating catastrophic. We report well-established Expected Calibration Error (ECE) (Guo et al., 2017) to assess the reliability of continual learning methods. More details are shown in the appendix.

**Training Details.** We adopt the standard experimental protocols following Boschini et al. (2022); Gu et al. (2023). All methods use a ResNet-18 backbone (He et al., 2016) trained from scratch with an SGD optimizer. For a fair comparison, we employ uniform settings across all methods (including epochs, batch sizes, and optimizer configurations). Datasets are split as follows: 5 tasks for CIFAR-10, and 10 tasks each for CIFAR-100 and TinyImageNet. We report the average results

Table 2: Comparison of Final Average Accuracy (FAA) across different continual learning methods on GCIL-CIFAR-100 dataset. All experiments are repeated 5 times with different seeds. Absolute gains are indicated in green.

| Method | Uniform | | | | Longtail | | | |
|---|---|---|---|---|---|---|---|---|
| | Buffer 200 | Δ | Buffer 500 | Δ | Buffer 200 | Δ | Buffer 500 | Δ |
| Joint (upper bound) | 58.36±1.02 | | | | 56.94±1.56 | | | |
| DER++ (Buzzega et al., 2020) | 19.36±0.65 | | 33.66±0.96 | | 27.05±1.11 | | 25.98±0.81 | |
| SCoMMER (Sarfraz et al., 2023) | 28.56±2.26 | | 35.70±0.86 | | 28.47±1.12 | | 32.99±0.49 | |
| ER (Riemer et al., 2019) | 16.34±0.74 | | 28.76±0.66 | | 19.55±0.69 | | 20.02±1.05 | |
| **Ours (ER+ID)** | 26.66±0.63 | **+10.32** | 40.54±0.46 | **+11.78** | 30.04±0.58 | **+10.49** | 35.92±0.35 | **+15.90** |
| CLS-ER (Arani et al., 2022) | 22.37±0.48 | | 36.80±0.34 | | 28.34±0.99 | | 28.35±0.72 | |
| **Ours (CLS-ER+ID)** | 31.17±1.62 | **+8.80** | 37.57±1.81 | **+0.77** | 34.08±0.45 | **+5.74** | 36.75±0.62 | **+8.40** |
| SARL (Sarfraz et al., 2025) | 36.20±0.46 | | 38.73±0.66 | | 34.13±1.07 | | 34.64±0.49 | |
| **Ours (SARL+ID)** | 36.45±0.37 | **+0.25** | 39.65±0.43 | **+0.92** | 35.04±0.54 | **+0.91** | 35.67±0.74 | **+1.03** |

over 5 independent runs with different random seeds to ensure statistical reliability. Comprehensive hyperparameter settings and further implementation details are provided in the appendix.

## 4.1 RESULTS

**Comparison with the state-of-the-art methods.** We evaluate our method against state-of-the-art continual learning approaches across three benchmark datasets with different memory buffer sizes: CIFAR-10, CIFAR-100, and Tiny-ImageNet. The Final Average Accuracies in the class incremental learning setting on different benchmarks are reported in Table 1. Our method outperforms all rehearsal-based methods on three datasets. Notably, our method outperforms the second best method BFP by up to 3% on CIFAR-10, which shows that our method remains highly effective even on a small-scale benchmark. Though outperforming XDER only slightly in FAA on CIFAR-100 and Tiny-ImageNet, our approach attains this accuracy with markedly lower computational cost, which can be shown in Figure 7 (a). Figure 6 shows that IDER has better performance at most intermediate tasks and also the final one. In addition, Table 2 highlights the advantage of IDER in the challenging GCIL setting, which tests the model's ability to deal with class imbalance and to continuously integrate knowledge from overlapping classes. The results in such a challenging setting prove the benefits of idempotence, which encourages the model to produce more robust representations to identify concepts clearly. This ability of IDER shows the potential for realistic continual learning.

**Plug-and-play with other rehearsal-based methods.** Considering the effectiveness and simplicity of idempotence, it is natural to consider whether it can be integrated into other rehearsal-based methods. Table 1 shows consistent performance improvements on various datasets with this integration. Enforcing idempotence boosts FAA by a significant margin, especially for ER ( 26% on CIFAR-10 with buffer size 200 and 21% on CIFAR-100 with buffer size 500). The results in GCIL in Table 2 can also prove that IDER, by enforcing model idempotence, is complementary to other methods in relieving forgetting. It is worth mentioning that in more challenging setting, the performance gains can be obvious in practice. Combined with CLS-ER, in traditional CIL, idempotence yields a gain of about 3.5% on CIFAR-100 with buffer size 500, while in GCIL, the gains can reach 8%. This additionally demonstrates the potential of this mathematical property to address catastrophic forgetting for even more challenging CL scenarios.

**Idempotence Improves prediction Reliability.** As previously reported by Guo et al. (2017), DNN are uncalibrated, often tending towards overconfidence. Arani et al. (2022) show that this problem is pronounced in continual learning where the models tend to be biased towards recent tasks. Following Boschini et al. (2022); Jha et al. (2023) we evaluate the calibration errors for different CL baselines using the well-established Expected Calibration Error (ECE), which is shown in Table 3. Table 3 shows that IDER consistently reduce the calibration error. In general, IDER benefits CL models in confidence calibration which demonstrates the ability of IDER to make reliable predictions. Comparing with post-hoc uncertainty calibration methods for CIL (Hwang et al., 2025; Li et al., 2024a), IDER achieves comparable performances across different datasets. This strong correlation between improved calibration and higher accuracy suggests that by producing more reliable confidence estimates, the model mitigates overconfidence on its own predictions (potentially incorrect), thereby facilitating a more stable and effective learning process that leads to better overall performance.

Table 3: Comparison of Expected Calibration Error (ECE) across different continual learning methods on CIFAR-10 and CIFAR-100 dataset. All experiments are repeated 5 times with different seeds. Results of NPCL are imported from its original work (Gu et al., 2023). Absolute improvements (lower ECE) are indicated in red.

| Method | CIFAR-10 | | | | CIFAR-100 | | | | Tiny-ImageNet | | | |
|---|---|---|---|---|---|---|---|---|---|---|---|---|
| | Buffer 200 | Δ | Buffer 500 | Δ | Buffer 500 | Δ | Buffer 2000 | Δ | Buffer 500 | Δ | Buffer 4000 | Δ |
| DER (Buzzega et al., 2020) | 29.91 | | 16.20 | | 24.84 | | 10.79 | | 22.80 | | 10.52 | |
| NPCL (Jha et al., 2023) | 21.03 | | - | | 19.95 | | - | | - | | - | |
| RC (Li et al., 2024a) | 16.39 | | 12.84 | | 19.43 | | 19.31 | | 21.32 | | 16.49 | |
| T-CIL (Hwang et al., 2025) | 22.50 | | 10.51 | | 15.79 | | 8.67 | | 14.50 | | 10.30 | |
| ER (Riemer et al., 2018) | 45.53 | | 32.69 | | 64.59 | | 45.64 | | 67.50 | | 51.37 | |
| **Ours (ER+ID)** | 12.36 | **-33.17** | 11.73 | **-20.96** | 13.65 | **-50.94** | 12.87 | **-32.77** | 21.55 | **-49.45** | 11.14 | **-40.23** |
| BFP (Gu et al., 2023) | 9.83 | | 9.40 | | 11.93 | | 9.28 | | 9.45 | | 8.25 | |
| **Ours (BFP+ID)** | 9.30 | **-0.53** | 8.63 | **-0.77** | 8.92 | **-3.01** | 8.29 | **-0.99** | 7.77 | **-1.68** | 6.35 | **-1.9** |

## 4.2 ADDITIONAL ANALYSIS

**Backbone Modification.** To enforce idempotence, we introduce a lightweight architectural modification by dividing the backbone into two parts (see Section 3.2) so that the model can incorporate the second input. We first evaluate whether this structural change alone affects performance on Split CIFAR-100 with a buffer size of 500. As shown in Table 4, the modified structure performs similarly to the normal backbone, indicating that the architectural change itself does not alter the baseline performance; therefore, the observed gains mainly come from the proposed idempotent loss rather than the modified architecture. We further ablate different partition points using ResNet-18 on Split CIFAR-100 and Tiny-ImageNet. Table 5 shows that shallower splits amplify noise and destabilize training, whereas deeper splits attenuate the second signal and weaken the idempotence effect. Based on these results, we split the backbone at a mid-layer, which yields the most consistent and strongest performance across different datasets overall.

Table 4: Comparison of performances using the normal vs. modified backbone. Similar results indicate that the architectural change itself does not affect baseline performance.

| Model | Method | Accuracy (%) | Forgetting (%) |
|---|---|---|---|
| Normal ResNet-18 | Finetune | 8.29 | 90.52 |
| | ER | 24.36 | 71.30 |
| Modified ResNet-18 | Finetune | 8.23 | 90.58 |
| | ER | 24.73 | 70.61 |

Table 5: Comparisons with different partition points on Split CIFAR-100 and Tiny-ImageNet using ResNet-18. Mid-layer partitioning yields the most stable and best performance.

| Layer Selection | CIFAR-100 | Tiny-ImageNet |
|---|---|---|
| Shallower (5th layer) | 40.51 | 42.10 |
| Deeper (13th layer) | 43.09 | 42.28 |
| **Ours (9th layer)** | **44.82** | **43.05** |

**Idempotence improves forgetting.** The Figure 7 (b) shows Final Forgetting (FF) measured on the Split CIFAR-100 dataset with different buffer sizes. Our method consistently reduces forgetting, which shows that enforcing idempotence improves accuracy while mitigating the forgetting problem.

**On training time.** Figure 7 (a) compares the training times of various methods. As expected, our proposed method introduces minimal computational overhead when integrated into existing replay-based methods. In practice, IDER only adds one extra forward pass, whereas X-DER involves additional training components (e.g., additional update steps beyond standard replay), leading to notably higher runtime.This highlights IDER's practicality as a lightweight and effective method.

**Comparison with different distance metrics.** The figure 7 (c) shows the effect of different distance metrics for computing the Idempotent distillation loss. While both MSE and KL divergence are well-established metrics for quantifying loss distance, MSE provides better performance. The reason is that MSE avoids the information loss occurring in probability space due to the squashing function.

**Visualizations of concrete example.** To explore how enforcing idempotence mitigates recency bias and improves calibration, we visualize predictions for test examples using ER and ER+ID on Split CIFAR-100 with a buffer size of 500 and 10 incremental tasks. Figure 8 shows top-5 predictions after learning the first two tasks on Split CIFAR-100. ER exhibits clear recency bias: classes from the current task (class ids from 10 to 19) receive inflated scores, leading to misclassifications. Integrated

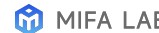

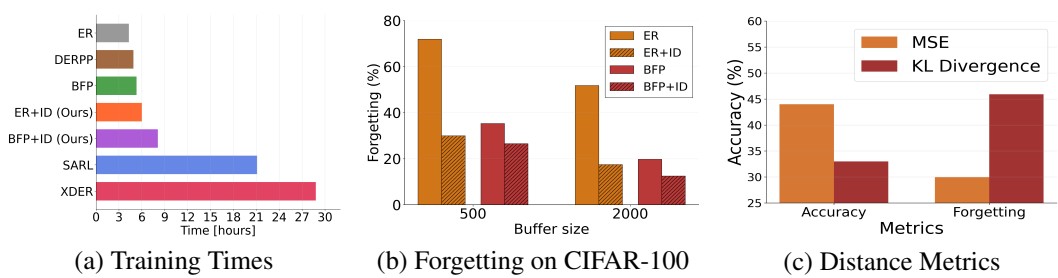

|                    |                          |                       |
| :----------------: | :----------------------: | :-------------------: |
| (a) Training Times | (b) Forgetting on CIFAR-100 | (c) Distance Metrics |

Figure 7: Results for model analysis. (a) the training time of different methods on Split TinyImageNet with buffer 500. (b) the Final Forgetting (FF) measures on Split CIFAR-100 with different buffer sizes. (c) the performances on Split CIFAR-100 using different distance metrics for idempotent distillation loss.

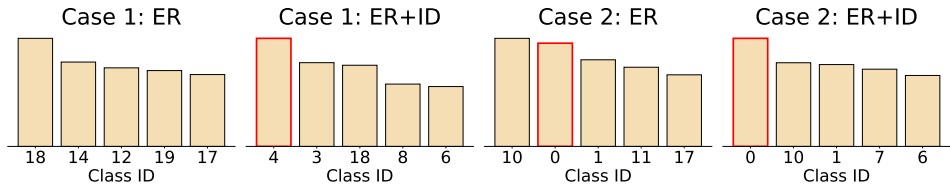

Figure 8: Visualizations of the predictions on Split CIFAR-100. It shows top-5 prediction probability produced by ER and ER+ID (ours) after training the first two tasks. The first 10 classes (class ids from 0 to 9) belong to task 1 and the next 10 classes (class ids from 10 to 19) belong to task 2. The ground-truth class is highlighted with red boxes.

with idempotence loss, the predictions get corrected as true class is promoted to top-1 prediction and overconfidence on new classes notably reduced.

**Cross-Platform Performance.** To assess the hardware sensitivity of our results, we run ER+ID with the same configuration on two different platforms: NVIDIA RTX 4090 and Huawei Ascend 910B. As Table 6 shows, the performance is consistent across platforms: the differences are small suggesting that the gains from IDER are not hardware-specific. Minor discrepancies are expected due to differences in kernels, numerical precision, and non-determinism in deep learning libraries.

Table 6: Performance comparison for ER+ID between NVIDIA RTX 4090 and Huawei Ascend 910B. All experiments are repeated 5 times with different seeds.

| Platform    | CIFAR-10   |            | CIFAR-100  |             | Tiny-ImageNet |
| :---------- | :--------: | :--------: | :--------: | :---------: | :-----------: |
|             | Buffer 200 | Buffer 500 | Buffer 500 | Buffer 2000 | Buffer 4000   |
| RTX 4090    | 71.02±1.98 | 74.74±0.42 | 44.82±0.85 | 56.59±0.35  | 43.05±1.40    |
| Ascend 910B | 71.13±0.92 | 74.68±0.37 | 44.87±0.76 | 56.44±0.41  | 43.01±1.26    |

## 5 CONCLUSION

In this paper, we propose Idempotent Experience Replay (IDER), a simple and effective method designed to mitigate catastrophic forgetting and improve predictive reliability in continual learning. Our approach adapts the training loss and introduces idempotence distillation loss for CL methods to encourage . Extensive experiments demonstrate that IDER consistently improves performance across multiple datasets and diverse continual learning settings. Our results show that enforcing idempotence enables a balance between stability and plasticity while yielding better calibrated predictions. Our method, requiring only two forward passes without additional parameters and seamlessly integrated with other CL approaches, shows promise for deployment of CL models in real-world scenarios. We hope this work inspires future research to place greater emphasis on uncertainty-aware continual learning. We also plan to explore the potential of idempotence property in different domains.

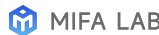

## ACKNOWLEDGEMENT

This project is supported by the National Natural Science Foundation of China (No. 62406192), the Shanghai Municipal Special Program for Basic Research on General AI Foundation Models (Grant No. 2025SHZDZX025G03), Opening Project of the State Key Laboratory of General Artificial Intelligence (No. SKLAGI2024OP12), the Tencent WeChat Rhino-Bird Focused Research Program, Kuaishou Technology, and the SJTU Kunpeng & Ascend Center of Excellence.

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

# Appendix

## A    EXTENDED DISCUSSION: RELATED IDEMPOTENCE TO CONTINUAL LEARNING

This section presents the theoretical motivation and formulation of the idempotence loss in continual learning by revisiting prior applications of idempotence and elaborating how the proposed loss improves prediction calibration and mitigates catastrophic forgetting

**Uncertainty Measurement**

Durasov et al. (2024a) first train the model to satisfy $f(x, 0) \approx y$ and $f(x, y) \approx y$ for each pair $(x, y)$ by minimizing the following loss:

$$L_{\text{train}} = \|f(x, 0) - y\| + \|f(x, y) - y\|. \tag{9}$$

Then they define the uncertainty loss as:

$$L(x) = \|y_1 - y_0\|, \tag{10}$$

where the network is applied recursively: $y_0 = f(x, 0)$, $y_1 = f(x, y_0)$.

The rationale is as follows:

1. If $x$ is in-distribution, then $y_0 \approx y$, and since the network is trained so that $f(x, y) \approx y$, they have $y_1 \approx y_0$. Therefore, the loss is small.

2. If $x$ is OOD, then $y_0$ is unlikely to approximate the true label. In this case, the pair $(x, y_0)$ is not a valid input as pretraining, leading $y_1$ to be unpredictable and significantly different from $y_0$, resulting in a large $\mathcal{L}(x)$.

Thus, the magnitude of $\|y_1 - y_0\|$ serves as a proxy for prediction certainty.

**Introducing Idempotence in Continual Learning**

In CL, models are often poorly calibrated and over-confident, a problem exacerbated by recency bias toward new tasks. To mitigate this issue, we require the model to maintain stable predictions on data from previous tasks even after parameter updates induced by new knowledge, as self-consistency indicates that the network's output is aligned with the learned in-distribution manifold and can make reliable (well-calibrated) prediction. As before, this condition can be translated into enforcing idempotence. We can formalize the desired idempotence condition as $f_t(x, f_t(x, 0)) = f_t(x, 0)$, where $x$ is from both previous and current tasks and $f_t$ represents the current model. In practice, as data from previous tasks can't be obtained, the loss can be defined as:

$$\mathcal{L} = \sum_{(x,y) \in \mathcal{T}_{t,M}} \|f_t(x, 0) - f_t(x, f_t(x, 0))\|_2^2, \tag{11}$$

where $M$ is the buffer memory which stores data from previous tasks.

Minimizing this loss drives the network toward the condition that repeated application of $f_t(x, \cdot)$ does not change the output, which is needed for model to make reliable predictions in CL.

However, minimizing the idempotence loss in CL is not trivial. First, we propose $\mathcal{L}_{ice}$ to train model idempotent for sequential tasks. Second, We modify the idempotence distillation loss by using the model checkpoint at the end of the last task for the second application, which can be rewritten as

$$\mathcal{L}_{\text{ide}} = \sum_{(x,y) \in \mathcal{T}_{t,M}} \|f_t(x, 0) - f_{t-1}(x, f_t(x, 0))\|_2^2. \tag{12}$$

The modification has two benefits:

- It prevents training collapse and bias error amplification. Consistent with Shocher et al. (2023) and Durasov et al. (2024c) , directly optimizing the idempotence loss induces two gradient pathways: 1. A desirable pathway that updates $f_t(x, 0)$ toward the correct in-distribution manifold. 2. An undesirable pathway that may cause the manifold to

expand, thereby including an incorrect $f_t(x, 0)$. For example, if $y_0 = f_t(x, 0)$ is an incorrect prediction, then minimizing $\|y_0 - y_1\|$ may cause $y_1 = f_t(x, y_0)$ to be pulled toward the incorrect $y_0$ and expand the manifold following the wrong gradient pathways, thereby magnifying the error. Another potential problem is to encourage $f_t(x, )$ to become the identity function, which is trivially idempotent and may cause training collapse. To counteract the latter gradient pathways, a frozen copy of the network is often used.

- It is designed for enforcing idempotence in CL and can serve as a distillation loss. According to eq 3, under empirical risk minimization, we can assume that:

$$f_t(x, f_t(x, \mathbf{0})) = f_t(x, \mathbf{0}). \tag{13}$$

Thus, we rewrite the $\mathcal{L}_{ide}$ as:

$$\mathcal{L}_{\text{ide}} = \sum_{(x,y) \in \mathcal{T}_{t,M}} \|f_t(x, f_t(x, 0)) - f_{t-1}(x, f_t(x, 0))\|_2^2. \tag{14}$$

First, according to the same input for $f_t$ and $f_{t-1}$ in eq 14, this idempotent distillation loss could serve as a standard regularization loss :

$$\mathcal{L}_{re} = \sum_{(x,y) \in \mathcal{T}_{t,M}} \|f_t(x) - f_{t-1}(x)\|_2^2. \tag{15}$$

which is often used in CL methods (Gu et al., 2023; Sarfraz et al., 2025) to mitigate catastrophic forgetting.

Second, when incorporating the second input that conveys logits from $f_t$, the loss steers the current model $f_t$ to update in a direction where the predictions remain correctly interpretable by the previous model $f_{t-1}$. Consequently, in sequential tasks, $f_{t-1}$ and $f_t$ are driven toward idempotence, feeding back $f_t$'s own output does not alter the prediction of $f_{t-1}$, yielding more reliable predictions across tasks and improving calibration in continual learning.

## B $t$-SNE VISUALIZATION OF VARIOUS METHODS

In this section, we show more $t$-SNE visualization of various methods on first task testing data on CIFAR-100 with 500 buffer size. The task number is 10. $t$-SNE figures shows our method has the better capability of resisting catstrophic forgetting compared with ER, DER and BFP. We observed that the feature clusters of the 10 classes from the first task become increasingly blurred as the model learns knowledge from new tasks in these methods. However, this phenomenon is alleviated in our method.

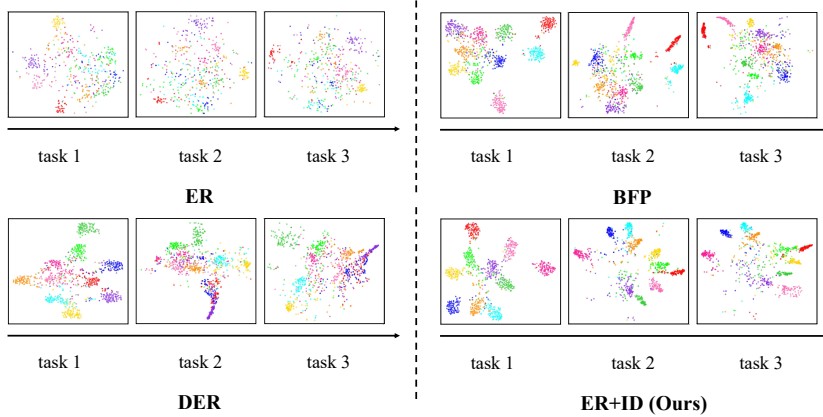

Figure 9: We perform $t$-SNE visualization of the features extracted from the first task testing data on CIFAR-100 across training tasks. The figures show how the feature clusters of the 10 classes from the first task change when the model train data from new tasks.

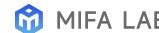

Table 7: Comparison of Final Forgetting (FF) across different continual learning methods. All experiments are repeated 5 times with different seeds. The best results (lowest forgetting) are highlighted in blue. The second best results are highlighted in green.

| Method | CIFAR-10 | | CIFAR-100 | | Tiny-ImageNet | |
|---|---|---|---|---|---|---|
| | Buffer 200 | Buffer 500 | Buffer 500 | Buffer 2000 | Buffer 500 | Buffer 4000 |
| Joint | 0.00 | 0.00 | 0.00 | 0.00 | 0.00 | 0.00 |
| iCaRL (Rebuffi et al., 2017) | 36.00±7.29 | 30.19±3.38 | 28.58±0.84 | 24.24±0.58 | 20.82±1.31 | 15.29±0.53 |
| ER (Riemer et al., 2018) | 71.35±7.77 | 52.12±7.56 | 71.92±0.74 | 51.82±0.75 | 74.79±0.67 | 57.47±0.57 |
| BiC (Wu et al., 2019) | 53.63±7.18 | 24.87±0.98 | 48.87±0.91 | 38.50±1.09 | 67.57±1.98 | 63.48±0.28 |
| LUCIR (Hou et al., 2019) | 59.79±10.16 | 37.58±3.80 | 50.22±1.26 | 32.48±0.76 | 35.02±0.56 | 30.59±0.95 |
| DER (Buzzega et al., 2020) | 45.31±2.73 | 32.04±2.88 | 56.66±2.55 | 34.41±2.05 | 68.43±2.73 | 59.54±6.13 |
| DER++ (Buzzega et al., 2020) | 36.20±4.15 | 27.93±3.34 | 51.85±1.61 | 34.44±1.42 | 61.45±3.12 | 39.11±3.66 |
| ER-ACE (Caccia et al., 2021) | 19.52±1.23 | 12.87±1.06 | 38.61±1.15 | 28.42±0.55 | 40.97±1.38 | 29.37±1.09 |
| XDER (Boschini et al., 2022) | 16.36±0.91 | 12.81±0.48 | 24.15±1.37 | 11.17±1.21 | 42.90±0.54 | 18.87±1.08 |
| BFP (Gu et al., 2023) | 22.53±5.00 | 16.81±1.11 | 35.32±3.94 | 19.76±0.87 | 30.02±4.37 | 27.19±6.53 |
| **Ours (ER+ID)** | 15.28±2.41 | 11.93±0.49 | 29.98±2.52 | 17.46±1.04 | 36.63±3.37 | 22.46±±1.86 |
| **Ours (BFP+ID)** | 15.79±2.73 | 12.11±0.82 | 26.56±2.56 | 12.52±1.29 | 27.41±3.92 | 16.68±0.44 |

## C  FORGETTING COMPARISON OF DIFFERENT REHEARSAL-BASED CONTINUAL LEARNING METHODS

Instead of FAA, Final Forgetting (FF) reflects the model's anti-forgetting capacity. To make fair comparison, we exclude the Exponential Moving Average (EMA) based methods, as the highest model performance on each task is from working model instead of EMA model, thereby reducing FF following Eq. 16 in appendix. FF measures the drop from each task's historical peak accuracy (its best accuracy when first learned) to its final accuracy after all tasks, while EMA artificially smooths performance drops by reducing the accuracy it is first learned, thus understating true forgetting. The Table 7 shows Final Forgetting (FF) of different continual learning methods. The results show that the idempotence loss yields a lower FF, indicating improved stability. It is worth noting that compared with XDER, there are no additional architectures or learnable parameters introduced in our method, just forwarding pass the model twice.

## D  EXPERIMENTAL SETTING

We evaluate our method on three standard continual learning benchmarks under Class-IL setting, where task identifiers are unavailable during testing, making it a challenging scenario for maintaining performance across tasks.

**Datasets.** Our experiments use three datasets with varying complexity:

- **Split CIFAR-10**: The CIFAR-10 dataset is divided into 5 sequential tasks, each containing 2 classes. Each class comprises 5,000 training and 1,000 test images of size 32×32.

- **Split CIFAR-100**: CIFAR-100 is split into 10 tasks with 10 classes per task. Each class contains 500 training and 100 test images of size 32×32.

- **Split TinyImageNet**: TinyImageNet is divided into 10 tasks with 20 classes each. Each class has 500 training images, 50 validation images, and 50 test images.

**Evaluation Metrics.** We use two standard metrics to evaluate continual learning performance:

- **Final Average Accuracy (FAA)**: Measures the average accuracy across all tasks after training is complete. For a model that has finished training on task $t$, let $a_i^t$ denote the test accuracy on task $i$. FAA is computed as the mean accuracy across all tasks.

- **Final Forgetting (FF)**: Quantifies how much knowledge of previous tasks is forgotten, defined as:

$$\text{FF} = \frac{1}{T-1} \sum_{i=1}^{T-1} \max_{j \in \{1, \cdots, T-1\}} (a_i^j - a_i^T). \tag{16}$$

where lower values indicate better retention of previously learned tasks.

- **Expected Calibration Error (ECE)**: Quantifies the mismatch between a model's predicted confidence and its actual accuracy. Predictions are partitioned into $M$ confidence interval bins $B_m$. The ECE is computed as the weighted average of the absolute difference between the average confidence ($\text{conf}(B_m)$) and the average accuracy ($\text{acc}(B_m)$) within each bin:

$$\text{ECE} = \sum_{m=1}^{M} \frac{|B_m|}{N} \left| \text{conf}(B_m) - \text{acc}(B_m) \right|. \tag{17}$$

where $N$ is the total number of samples. A lower ECE indicates a better-calibrated model whose confidence estimates are more reliable.

### D.1 IMPLEMENTATION DETAILS SUPPLEMENTARY

Besides the details mentioned above, we train 50 epochs per task for Split CIFAR-10 and Split CIFAR-100 and 100 epochs per task for Split TinyImageNet (Le & Yang, 2015). For Split CIFAR100, the learning rate is decreased by a factor of 0.1 at epochs 35 and 45, while for Split TinyImageNet,the learning rate is decreased by a factor of 0.1 at epochs 35, 60 and 75. The learning rate may vary in the light of different continual learning methods, while for a fair comparison, we use the same initial learning rate as DER and BFP for our methods. If not specified, all baselines use the reservoir sampling algorithm (Vitter, 1985) to update memory, while BFP (Gu et al., 2023) uses class-balanced reservoir sampling (Buzzega et al., 2021) for pushing balanced examples into the buffer.

### D.2 HYPERPARAMETERS

In this section, we show hyperparameter combination that used in our experiments. These hyperparameters are adopted from Boschini et al. (2022); Buzzega et al. (2020); Gu et al. (2023) to make fair comparison.

SPLIT CIFAR-10

**Buffer size = 200**

> **iCaRL:** $lr = 0.1,\ wd = 10^{-5}$
> **LUCIR:** $\lambda_{\text{base}} = 5$, mom $= 0.9$, $k = 2$, epoch$_{\text{fitting}} = 20$, $lr = 0.03$, $lr_{\text{fitting}} = 0.01$, $m = 0.5$
> **BiC:** $\tau = 2$, epochs$_{\text{BiC}} = 250$, $lr = 0.03$
> **ER-ACE:** $lr = 0.03$
> **ER:** $lr = 0.1$
> **DER:** $lr = 0.03,\ \alpha = 0.3$
> **DER++:** $lr = 0.03,\ \alpha = 0.1,\ \beta = 0.5$
> **XDER:** $\alpha = 0.3,\ m = 0.7,\ \beta = 0.9,\ \gamma = 0.85,\ wd = 1e-06,\ \lambda = 0.05,\ \eta = 0.001,\ lr = 0.03,\ \tau = 5,\ mom = 0.9$
> **DEP++ w/ BFP:** $lr = 0.03,\ \alpha_{distill} = 0.1,\ \alpha_{ce} = 0.5,\ \alpha_{bfp} = 1$

**Buffer size = 500**

> **iCaRL:** $lr = 0.1,\ wd = 10^{-5}$
> **LUCIR:** $\lambda_{\text{base}} = 5$, mom $= 0.9$, $k = 2$, epoch$_{\text{fitting}} = 20$, $lr = 0.03$, $lr_{\text{fitting}} = 0.01$, $m = 0.5$

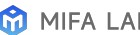

**BiC:** $\tau = 2$, $\text{epochs}_{\text{BiC}} = 250$, $lr = 0.03$
**ER-ACE:** $lr = 0.03$
**ER:** $lr = 0.1$
**DER:** $lr = 0.03$, $\alpha = 0.3$
**DER++:** $lr = 0.03$, $\alpha = 0.1$, $\beta = 0.5$
**XDER:** $\alpha = 0.3$, $m = 0.7$, $\beta = 0.9$, $\gamma = 0.85$, $wd = 1e - 06$, $\lambda = 0.0$, $\eta = 0.001$, $lr = 0.03$, $\tau = 5$, $mom = 0.9$
**DEP++ w/ BFP:** $lr = 0.03$, $\alpha_{distill} = 0.2$, $\alpha_{ce} = 0.5$, $\alpha_{bfp} = 1$

SPLIT CIFAR-100

**Buffer size = 500**

**iCaRL:** $lr = 0.3$, $wd = 10^{-5}$
**LUCIR:** $\lambda_{\text{base}} = 5$, $mom = 0.9$, $k = 2$, $\text{epoch}_{\text{fitting}} = 20$, $lr = 0.03$, $lr_{\text{fitting}} = 0.01$, $m = 0.5$
**BiC:** $\tau = 2$, $\text{epochs}_{\text{BiC}} = 250$, $lr = 0.03$
**ER-ACE:** $lr = 0.03$
**ER:** $lr = 0.1$
**DER:** $lr = 0.03$, $\alpha = 0.3$
**DER++:** $lr = 0.03$, $\alpha = 0.1$, $\beta = 0.5$
**XDER:** $\alpha = 0.3$, $m = 0.7$, $\beta = 0.9$, $\gamma = 0.85$, $wd = 1e - 06$, $\lambda = 0.05$, $\eta = 0.001$, $lr = 0.03$, $\tau = 5$, $mom = 0.9$
**DEP++ w/ BFP:** $lr = 0.03$, $\alpha_{distill} = 0.1$, $\alpha_{ce} = 0.5$, $\alpha_{bfp} = 1$

**Buffer size = 2000**

**iCaRL:** $lr = 0.3$, $wd = 10^{-5}$
**LUCIR:** $\lambda_{\text{base}} = 5$, $mom = 0.9$, $k = 2$, $\text{epoch}_{\text{fitting}} = 20$, $lr = 0.03$, $lr_{\text{fitting}} = 0.01$, $m = 0.5$
**BiC:** $\tau = 2$, $\text{epochs}_{\text{BiC}} = 250$, $lr = 0.03$
**ER-ACE:** $lr = 0.03$
**ER:** $lr = 0.1$
**DER:** $lr = 0.03$, $\alpha = 0.3$
**DER++:** $lr = 0.03$, $\alpha = 0.1$, $\beta = 0.5$
**XDER:** $\alpha = 0.3$, $m = 0.7$, $\beta = 0.9$, $\gamma = 0.85$, $wd = 1e - 06$, $\lambda = 0.05$, $\eta = 0.001$, $lr = 0.03$, $\tau = 5$, $mom = 0.9$
**DEP++ w/ BFP:** $lr = 0.03$, $\alpha_{distill} = 0.1$, $\alpha_{ce} = 0.5$, $\alpha_{bfp} = 1$

SPLIT TINYIMAGENET

**Buffer size = 4000**

**iCaRL:** $lr = 0.03$, $wd = 10^{-5}$
**LUCIR:** $\lambda_{\text{base}} = 5$, $mom = 0.9$, $k = 2$, $\text{epoch}_{\text{fitting}} = 20$, $lr = 0.03$, $lr_{\text{fitting}} = 0.01$, $m = 0.5$
**BiC:** $\tau = 2$, $\text{epochs}_{\text{BiC}} = 250$, $lr = 0.03$
**ER-ACE:** $lr = 0.03$
**ER:** $lr = 0.1$
**DER:** $lr = 0.03$, $\alpha = 0.1$
**DER++:** $lr = 0.03$, $\alpha = 0.1$, $\beta = 0.5$

> **XDER:** $\alpha = 0.3, m = 0.7, \beta = 0.9, \gamma = 0.85, wd = 1e - 06, \lambda = 0.0, \eta = 0.001, lr = 0.03, \tau = 5, mom = 0.9$
>
> **DEP++ w/ BFP:** $lr = 0.03, \alpha_{distill} = 0.3, \alpha_{ce} = 0.8, \alpha_{bfp} = 1$

### D.3 INTERGRATED IDER INTO BFP

As Gu et al. (2023) introduces BFP distillation loss, which focuses on features. We can easily incorporate our method into BFP framework. The BFP loss is:

$$\mathcal{L}_{BFP} = \sum_{(x,y) \in \mathcal{T}_t, M} \|A h_t(x, 0) - h_{t-1}(x, 0)\|_2, \tag{18}$$

where $h_t$ is feature extractor in model $f_t$ on the t-th task and $A$ is linear transformation aimed to preserve the linear separability of features backward in time.

During training on the task $t$, the model $f_t$ and $A$ are optimized respectively. Thus, the $\mathcal{L}_{BFP+ID}$ can be:

$$\mathcal{L}_{BFP+ID} = \mathcal{L}_{ice} + \alpha \mathcal{L}_{ide} + \beta \mathcal{L}_{rep\text{-}ice} + \gamma \mathcal{L}_{BFP}. \tag{19}$$

### D.4 COMPLEXITY AND TRAINING COST

We train all experiments on NVIDIA RTX 4090 and Ascend 910B. The additional parameters we need for modified architecture are very small. Using ResNet-18 as the backbone, the normal architecture contains 11.22M parameters on CIFAR-100 while the modified architecture increases the parameter count to 11.91M parameters. Although we need two forward passes to train the model, which leads to a slightly longer training time compared with DER++, the longer training time is acceptable as it increases the performance by a significant margin.

## E LIMITATIONS

**Naive Implementation.** As we first introduce idempotence property into continual learning, the method should be very simple. In the future, we try to combine our method with more complementary techniques to further improve performance. In addition, we also plan to explore the application of the idempotence property in data sampling strategies for continual learning.

## F ABLATION RESULTS

### F.1 CONTRIBUTION OF EACH COMPONENT

We conduct a component ablation study to isolate the contributions of each part of the overall objective: the Standard Idempotent Module (SIM), the Idempotent Distillation Module (IDM), and Experience Replay (ER). We focus on the ER+ID method for this study. The ablation study results are shown in Table 8. Using only the Standard Idempotent Module (SIM) or combing SIM with ER produces similar performance compared with finetuning or ER baseline, indicating that SIM alone trains model to be idempotent on the current task, which making it well-suited for subsequent idempotent distillation and doesn't mitigate catastrophic forgetting. It also demonstrates that modified architecture does not influence performance and the observed improvements benefit from idempotent distillation loss. Whta's more, adding the Idempotent Distillation Module (IDM) yields substantial performance gains, which further improves accuracy to 44.82%. This validates the effectiveness of idempotence distillation loss.

Table 8: Ablation study of different components on Split CIFAR-100.

| SIM | IDM | ER | FAA |
|-----|-----|-----|-----|
| ✓ | ✗ | ✗ | 8.23 |
| ✓ | ✗ | ✓ | 24.73 |
| ✓ | ✓ | ✓ | 44.82 |

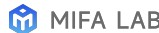

### F.2 HYPERPARAMETER SENSITIVITY

We performed ablations on CIFAR-100 with 500 buffer size under CIL setting. We ablate on ER+ID method, as it consistently yields substantial improvements over ER baseline across all datasets and does not introduce any additional hyperparameters. Both $\alpha$ and $\beta$ are from the set $\{0.1, 0.2, 0.5, 1\}$. Table 9 demonstrates that IDER is not overly sensitive to specific hyperparameter values, as multiple configurations yield consistent performance improvements. This reliability underscores IDER's practicality and suitability for a wide range of continual learning applications.

Table 9: Impact of hyperparameters $\alpha$ and $\beta$ for ER+ID on Final Average Accuracy (FAA) on Split CIFAR-100. We use learning rate as 0.03 and training epoch as 50.

| beta | alpha=0.1 | alpha=0.2 | alpha=0.5 | alpha=1 |
|------|-----------|-----------|-----------|---------|
| 0.1  | 44.77     | 44.24     | 41.74     | 40.42   |
| 0.2  | 44.41     | 44.43     | 44.25     | 43.02   |
| 0.5  | 43.82     | 44.33     | **44.82** | 43.87   |
| 1    | 41.20     | 41.58     | 42.72     | 40.93   |

### F.3 PROBABILITY SELECTION

We provide ablation study of the probability $P$ used in the Standard Idempotent Module, which determines whether the second input is set to the empty signal or the ground-truth one-hot vector. We perform a sensitivity ablation on Split CIFAR-100 with 500 buffer size and on ER+ID method. In our ablation study, the second input is set to the neutral "empty" signal input 0 with probability $P$ and the ground-truth one-hot vector y with probability $1 - P$. Table 10 shows that as $P$ increases, FAA consistently improves and peaks at $P = 0.9$, which reaches 44.82. Then there is a slight drop at $P = 1.0$, where FAA is 43.26. Therefore, we use $P = 0.9$ by default. For simplicity, we can also set $P = 1.0$, which yields comparable performance.

Table 10: Impact of the probability $P$ on Final Average Accuracy (FAA) on Split CIFAR-100.

| P   | 0.2  | 0.3   | 0.4   | 0.5   | 0.6   | 0.7   | 0.8   | 0.9       | 1     |
|-----|------|-------|-------|-------|-------|-------|-------|-----------|-------|
| FAA | 20.9 | 22.24 | 27.03 | 29.19 | 31.46 | 37.48 | 42.24 | **44.82** | 43.26 |

## G STATEMENT ON THE USE OF LARGE LANGUAGE MODELS

We hereby declare that large language models (LLMs), specifically GPT-5 , were used during the preparation of this manuscript. The use of LLMs was strictly limited to: aiding and polishing writing. The LLM was used solely as an assistive tool for prose refinement and did not contribute to the intellectual content of the research.

