# OpenReview forum: "IDER: IDempotent Experience Replay for Reliable Continual Learning"
_ICLR.cc/2026/Conference — ICLR 2026 Poster_

### Official Review · Reviewer_WbzD · 2025-10-27

**Soundness:** 2
**Presentation:** 2
**Contribution:** 2
**Rating:** 2
**Confidence:** 4

**Summary:**

This paper introduces Idempotent Experience Replay (IDER), a method for class-incremental learning (CIL) that exploits the mathematical property of idempotence to improve accuracy, forgetting, and uncertainty calibration. The method consists of two components: 1) a standard idempotent module that trains the current model to be idempotent on the current task; and 2) an idempotent distillation module that enforces idempotence between the previous and current model checkpoints. Experiments on standard datasets demonstrate improvements in both accuracy and calibration.


**Recommendation**
I lean toward rejecting the paper in its current form. While the core idea of applying idempotence to continual learning is novel and the empirical results are encouraging, the paper suffers from significant theoretical and experimental gaps which prevent a rigorous and clear assessment of the paper.

**Strengths:**

1. **Novel application of idempotence to continual learning:** while idempotence has been explored in deep learning for other tasks, this appears largely under-explored in continual learning settings.
2. **Plug-and-play design with rehearsal-based approaches:** the method can be integrated with existing rehearsal-based approaches (e.g., ER) with minimal overhead, making it practically appealing in those cases.
3. **Experimental evaluation on general-CIL:** the method has also been tested on more realistic conditions (i.e., general class-incremental learning), making it interesting from a practical point of view.

**Weaknesses:**

1. **Limited theoretical justification:** while the paper takes inspiration from prior work on idempotence, the theoretical connection between idempotence and mitigation of both catastrophic forgetting and uncertainty miscalibration is not clear. Although being demonstrated empirically, it is unclear why enforcing idempotence specifically helps with continual learning problems in the first place. The intuition lacks formal analysis. Can you provide explanations on why idempotence should help with catastrophic forgetting and uncertainty calibration?
2. **Equation 6:** the idempotent distillation loss uses $f_{t-1}(x, f_t(x, 0))$ rather than $f_t(x, f_t(x, 0))$ to avoid pulling predictions toward incorrect output, but this breaks the true idempotence property since $f_{t-1}$ and $f_t$ are two different functions. What happens if you use $f_t(x, f_t(x, 0))$ instead of $f_{t-1}(x, f_t(x, 0))$ in Equation 6?
3. **Missing evaluation with calibration methods for CL:** one of the main focus of the paper is reliability (i.e., reducing the calibration error). However, the paper does not consider recent work on uncertainty calibration for class-incremental learning [1,2].
4. **Limited comparison with other CL methods:** apart from rehearsal-based methods, it would be interesting to include comparison with, e.g., regularisation-based or parameter-isolation approaches to demonstrate broader applicability.
5. **Incomplete experimental analysis:**
    1. Hyperparameter sensitivity: no analysis of sensitivity to $\alpha$ and $\beta$ introduced in Equation 8.
    2. Probability $P$: the probability is not reported nor ablated.
    3. Backbone architecture: only ResNet18 is tested. What will happen when using a different architecture? Again, this would demonstrate broader applicability.
    4. Limited experiments on GCIL (Table 2): it would be interesting to see the results on the other considered datasets.
    5. Limited calibration results (Table 3): as anticipated above, the paper is missing recent work on calibration in continual learning. Furthermore, why is Tiny-ImageNet not included? Finally, since the results from NPCL were copied from the original work, do you assure that the experimental setting is exactly the same? Otherwise, the comparison is not meaningful.

**Minor issues**
1. No analysis on larger task sequences (e.g., 20 tasks on Tiny-ImageNet).
2. Figures 3, 4, and 5 are difficult to parse.
3. Writing style can be overall improved.

[1] Li, Lanpei, et al. "Calibration of continual learning models." Proceedings of the IEEE/CVF Conference on Computer Vision and Pattern Recognition. 2024.

[2] Hwang, Seong-Hyeon, Minsu Kim, and Steven Euijong Whang. "T-CIL: Temperature Scaling using Adversarial Perturbation for Calibration in Class-Incremental Learning." Proceedings of the Computer Vision and Pattern Recognition Conference. 2025.

**Questions:**

See above.

---

> ### Author Response · Authors · 2025-11-23
> **Response to Reviewer WbzD （1/6）**
>
> Thanks a lot for your review and detailed feedback! We would like to address your concerns below.
>
> ---
> **Q1**: Limited theoretical justification: while the paper takes inspiration from prior work on idempotence, the theoretical connection between idempotence and mitigation of both catastrophic forgetting and uncertainty miscalibration is not clear. Although being demonstrated empirically, it is unclear why enforcing idempotence specifically helps with continual learning problems in the first place. The intuition lacks formal analysis. Can you provide explanations on why idempotence should help with catastrophic forgetting and uncertainty calibration?
>
> **A1**: Thank you for the valuable feedback. Below, we provide detailed explanations on why idempotence can help continual learning resist catastrophic forgetting and improve uncertainty calibration. More details can be seen in appendix F.
>
> In CL, models are often poorly calibrated and over-confident, a problem exacerbated by recency bias toward new tasks. To mitigate this issue, we require **the model to maintain stable predictions on data from previous tasks even after parameter updates induced by new knowledge, as self-consistency indicates that the network's output is aligned with the learned in-distribution manifold and can make reliable (well-calibrated) prediction. This condition can be translated into enforcing idempotence.** We can formalize the desired idempotence condition as $f_t(x, f_t(x, 0)) = f_t(x, 0)$, where $x$ is from both previous and current tasks and $f_t$ represents the current model. In practice, as data from previous tasks can't be obtained, the loss can be defined as:
>
> $$\mathcal{L} = \sum_{(x,y)\in \mathcal{T}_{t,M}} \left\lVert f_t(x, 0) - f_t(x, f_t(x, 0)) \right\rVert_2^2,$$
>
> where $M$ is the buffer memory which stores data from previous tasks.
>
> Minimizing this loss drives the network toward the condition which is needed for model to make reliable predictions in CL. However, minimizing the idempotence loss in CL is not trivial. First, we propose $\mathcal{L}_{\text{ice}}$ to train model idempotent for sequential tasks. Second, we modify the idempotence distillation loss by using the model checkpoint at the end of the last task for the second application, which can be rewritten as
>
> $$\mathcal{L}&#95;{\text{ide}} = \sum&#95;{(x,y)\in \mathcal{T}&#95;{t,M}} \left\lVert f&#95;t(x, 0) - f&#95;{t-1}(x, f&#95;t(x, 0)) \right\rVert&#95;2^2.$$
>
> The modification has two benefits:
>
> **First, it prevents training collapse and bias error amplification.** Consistent with [1] and [2], directly optimizing the idempotence loss induces two gradient pathways: (1) A desirable pathway that updates $f_{t}(x, 0)$ toward the correct in-distribution manifold. (2) An undesirable pathway that may cause the manifold to expand, thereby including an incorrect $f_{t}(x, 0)$.  **To counteract the latter gradient pathways, a frozen copy of the network is often used.**
>
> **Second, it is designed for enforcing idempotence in CL and can serve as a distillation loss.** According to $\mathcal{L}_{\text{ice}}$, under empirical risk minimization, we can assume that:
>
>   $$f_t(x, f_t(x, \mathbf{0})) = f_t(x, \mathbf{0}).$$
>
>   Thus, we rewrite the $\mathcal{L}_{\text{ide}}$ as:
>
> $$\mathcal{L}&#95;{\text{ide}} = \sum&#95;{(x,y)\in \mathcal{T}&#95;{t,M}} \left\lVert f&#95;t(x,f&#95;t(x, 0)) - f&#95;{t-1}(x, f&#95;t(x, 0)) \right\rVert&#95;2^2.$$
>
>   **First, according to the same input for $f_t$ and $f_{t-1}$ in this equation, this idempotent distillation loss could serve as a standard regularization loss**:
>
> $$\mathcal{L}&#95;{\text{re}} = \sum&#95;{(x,y)\in \mathcal{T}&#95;{t,M}} \left\lVert f&#95;t(x) - f&#95;{t-1}(x) \right\rVert&#95;2^2,$$
>
>   **which is often used in CL methods[3,4] to mitigate catastrophic forgetting**. Second, when incorporating the second input that conveys logits from $f_t$, the loss steers the current model $f_t$ to update in a direction where the predictions remain correctly interpretable by the previous model $f_{t-1}$. **Consequently, in sequential tasks, $f_{t-1}$ and $f_t$ are driven toward idempotence, feeding back $f_t$'s own output does not alter the prediction of $f_{t-1}$, yielding more reliable predictions on data from previous tasks to improve calibration in continual learning**.
>
> [1] Shocher, Assaf, et al. "Idempotent generative network." In The Thirteenth International Conference on Learning Representations, 2024.
>
> [2] Durasov, Nikita, et al. "IT³: Idempotent Test-Time Training." ICML 2025.
>
> [3] Gu, Qiao, Dongsub Shim, and Florian Shkurti. "Preserving linear separability in continual learning by backward feature projection." Proceedings of the IEEE/CVF Conference on Computer Vision and Pattern Recognition. 2023.
>
> [4] Sarfraz, Fahad, Elahe Arani, and Bahram Zonooz. "Semantic Aware Representation Learning for Lifelong Learning." The Thirteenth International Conference on Learning Representations.

---

> ### Author Response · Authors · 2025-11-23
> **Response to Reviewer WbzD （2/6）**
>
> **Q2**: Equation 6: the idempotent distillation loss uses $f_{t-1}(x, f_t(x, 0))$ rather than $f_t(x, f_t(x, 0))$ to avoid pulling predictions toward incorrect output, but this breaks the true idempotence property since $f_{t-1}$ and $f_t$ are two different functions. What happens if you use $f_t(x, f_t(x, 0))$ instead of $f_{t-1}(x, f_t(x, 0))$ in Equation 6?
>
> **A2**:  Thank you for your question. **We intentionally use a frozen checkpoint at the end of the last task $f_{t-1}$ in the second pass to enforce idempotence.** The more details are explained in A1 and appendix F.
>
> **First, using the frozen model instead of obeying true idempotence property is widely used in [1,2], which prevents training collapse and bias error amplification.**
>
> **Second, it is designed for enforcing idempotence in CL and can serve as a distillation loss.**
>
> Third, even though parameters may be different, **both $f_{t}$ and $f_{t−1}$ are the same model at different time steps, which satisfies the idempotence for the same model and is more suitable given the goal of CL**. **Our experiment justifies this.** Empirically, replacing $f_{t−1}$ with $f_{t}$  degrades the continual learning performance as the FAA drops 6% on split CIFAR100 with 500 buffer size.
>
> |  | FAA |
> |-----------|--------|
> | $f_{t-1}(x, f_t(x, 0))$ | 44.82 |
> | $f_t(x, f_t(x, 0))$ | 38.57 |
>
>
> [1] Shocher, Assaf, et al. "Idempotent generative network." In The Thirteenth International Conference on Learning Representations, 2024.
>
> [2] Durasov, Nikita, et al. "IT³: Idempotent Test-Time Training." ICML 2025.
>
>
> ---
> **Q3**: Missing evaluation with calibration methods for CL: one of the main focus of the paper is reliability (i.e., reducing the calibration error). However, the paper does not consider recent work on uncertainty calibration for class-incremental learning [1,2].
>
> **A3**: Thanks for your suggestion. Our work aims to introduce idempotence during training for rehearsal-based continual learning, both improving reliability (better-calibrated predictions) and mitigating catastrophic forgetting.  **While our approach may appear close to [1,2], there is a key difference:  they [1,2]  focus on post-hoc adjustments after training, which require a separate validation set and typically not enhancing CL metrics such as final average accuracy or forgetting.**
>
> To further evaluate uncertainty calibration for class-incremental learning, we still have added comparisons against [1,2] under matched settings. The results are shown below. **Our method can have comparable performance on reducing the calibration error. Combined with BFP, our method can achieve the best performance,** underscoring robustness of our method for reliable continual learning.
> | Method                        | Tinyimg |       | CIFAR-100 |       | CIFAR-10 |       |
> |:----------------------:|:-------:|:-----:|:---------:|:-----:|:--------:|:-----:|
> |                        | 500     | 4000  | 500       | 2000  | 200      | 500   |
> | DER                    | 22.80   | 10.52 | 24.84     | 10.79 | 29.91    | 16.20 |
> | Continual Calibration  | 21.32   | 16.49 | 19.43     | 19.31 | 16.39    | 12.84 |
> | T-CIL                  | 14.50   | 10.30 | 15.79     | 8.67  | 22.50    | 10.51 |
> | ER                     | 67.50   | 51.37 | 64.59     | 45.64 | 45.53    | 32.69 |
> | ER+ID (Ours)          | 21.55   | 11.14 | 13.65     | 12.87 | 12.36    | 11.73 |
> | BFP                    | 9.45    | 8.25  | 11.93     | 9.28  | 9.83     | 9.40  |
> | BFP+ID (Ours)         | 7.77    | 6.35  | 8.92      | 8.29  | 9.30     | 8.63  |
>
> [1] Li, Lanpei, et al. "Calibration of continual learning models." Proceedings of the IEEE/CVF Conference on Computer Vision and Pattern Recognition. 2024.
>
> [2] Hwang, Seong-Hyeon, Minsu Kim, and Steven Euijong Whang. "T-CIL: Temperature Scaling using Adversarial Perturbation for Calibration in Class-Incremental Learning." Proceedings of the Computer Vision and Pattern Recognition Conference. 2025.

---

> ### Author Response · Authors · 2025-11-23
> **Response to Reviewer WbzD （3/6）**
>
> **Q4:** Limited comparison with other CL methods: apart from rehearsal-based methods, it would be interesting to include comparison with, e.g., regularization-based or parameter-isolation approaches to demonstrate broader applicability.
>
> **A4:** Thank you for your valuable suggestion regarding the comparison with a broader range of continual learning (CL) methods, including regularization-based and parameter-isolation approaches. We appreciate your insights, as they help in assessing the applicability and robustness of our proposed method.
>
> First, it is important to clarify that **comparing parameter-isolation approaches with rehearsal-based methods is not fair.**  Parameter-isolation approaches often depend on **architecture expansion** to learn new knowledge while rehearsal-based methods usually retain a fixed architecture, making direct comparisons sensitive to these structural differences. In addition, most recent parameter-isolation approaches[1,2] depend on **pretrained backbones**. These pre-trained backbones often include representations of a significant portion of the classes encountered later in CL tasks or classes that are highly similar, which **provides prior advantages unrelated to the CL mechanism itself**. As a result, in comparison to rehearsal-based methods, the parameter-isolation methods have significant advantages. What's more, parameter-isolation methods are evaluated **under different experimental settings, further undermining direct comparisons.**
>
> Second, **we compare our method against some state-of-art regularization-based methods under the same experiment setting as ours**: LwF+NCM[3], LwF+SDC[4], PASS[5], FeTrIL[6], FeCAM[7], EFC[8], LwF+LDC[9]. For our methods, the buffer size is set to 2000 on CIFAR-100 and 4000 on Tiny-ImageNet. **The results below show that our methods generally achieve better performance.**
> | Method      | CIFAR100       | Tiny-ImageNet  |
> |:-----------:|:--------------:|:--------------:|
> | LwF+NCM     | 40.5±2.7       | 28.6±1.1       |
> | LwF+SDC     | 40.6±1.8       | 29.5±0.8       |
> | PASS        | 37.8±0.2       | 31.2±0.4       |
> | FeTrIL      | 37.0±0.6       | 24.4±0.6       |
> | FeCAM       | 33.1±0.9       | 24.9±0.5       |
> | EFC         | 43.6±0.7       | 34.1±0.8       |
> | LwF+LDC     | 45.4 ± 2.8       | 34.2±0.7       |
> | ER+ID       | 56.59±0.35     | 43.05±1.40     |
> | BFP+ID      | 57.74±0.64     | 43.51±0.59     |
> | CLS-ER+ID   | 56.36±0.78     | 46.17±0.22     |
>
> [1] Liang, Yan-Shuo, and Wu-Jun Li. "Inflora: Interference-free low-rank adaptation for continual learning." Proceedings of the IEEE/CVF Conference on Computer Vision and Pattern Recognition. 2024.
>
> [2]Zhou, Da-Wei, et al. "Expandable subspace ensemble for pre-trained model-based class-incremental learning." Proceedings of the IEEE/CVF Conference on Computer Vision and Pattern Recognition. 2024.
>
> [3]Rebuffi, S.A., Kolesnikov, A., Sperl, G., Lampert, C.H.: icarl: Incremental classifier and representation learning. In: Conference on Computer Vision and Pattern Recognition
>
> [4]Yu, L., Twardowski, Bartlomiej, Liu, Xialei, Herranz, L., Wang, K., Jui, S., Weijer, J.v.d.: Semantic drift compensation for class-incremental learning. In: Proceedings of the IEEE/CVF Conference on Computer Vision and Pattern Recognition. pp. 6982–6991 (2020)
>
> [5]Zhu, F., Zhang, X.Y., Wang, C., Yin, F., Liu, C.L.: Prototype augmentation and self-supervision for incremental learning. In: Proceedings of the IEEE/CVF Conference on Computer Vision and Pattern Recognition. pp. 5871–5880 (2021)
>
> [6]Petit, G., Popescu, A., Schindler, H., Picard, D., Delezoide, B.: Fetril: Feature translation for exemplar-free class-incremental learning. In: Proceedings of the IEEE/CVF Winter Conference on Applications of Computer Vision. pp. 3911– 3920 (2023)
>
> [7]Goswami, D., Liu, Y., Twardowski, B., van de Weijer, J.: Fecam: Exploiting the heterogeneity of class distributions in exemplar-free continual learning. Advances in Neural Information Processing Systems 36 (2024)
>
> [8]Magistri, S., Trinci, T., Soutif-Cormerais, A., van de Weijer, J., Bagdanov, A.D.: Elastic feature consolidation for cold start exemplar-free incremental learning. In: International Conference on Learning Representations (2024)
>
> [9]Gomez-Villa, Alex, et al. "Exemplar-free continual representation learning via learnable drift compensation." European Conference on Computer Vision. Cham: Springer Nature Switzerland, 2024.

---

> ### Author Response · Authors · 2025-11-24
> **Response to Reviewer WbzD （4/6）**
>
> **Q5**: Hyperparameter sensitivity: no analysis of sensitivity to $\alpha$ and $\beta$ introduced in Equation 8.
>
> **A5**: We appreciate the reviewers for identifying the lack of $\alpha$ and $\beta$ hyperparameter sensitivity analysis and we now have conducted ablation study on the $\alpha$ and $\beta$ introduced in Equation 8. **We focus on the ER+ID method for this study, as it consistently yields substantial improvements over ER baseline across all datasets and does not introduce any additional hyperparameters.** We conduct tests on the Split CIFAR-100 with a buffer size of 500. The ablation study results are shown below. **Our choice of hyperparameters in the paper yields optimal performance, and our methods demonstrates robustness to these hyperparameter selections.** We hope that these ablation studies will provide the reviewer with a more comprehensive understanding of our work.
> | alpha | beta | FAA   |
> |-------|------|-------|
> | 0.1   | 0.1  | 44.77 |
> |       | 0.2  | 44.41 |
> |       | 0.5  | 43.82 |
> |       | 1.0  | 41.20 |
> | 0.2   | 0.1  | 44.24 |
> |       | 0.2  | 44.43 |
> |       | 0.5  | 44.33 |
> |       | 1.0  | 41.58 |
> | 0.5   | 0.1  | 41.74 |
> |       | 0.2  | 44.25 |
> |       | 0.5  | 44.82 |
> |       | 1.0  | 42.72 |
> | 1.0   | 0.1  | 40.42 |
> |       | 0.2  | 43.02 |
> |       | 0.5  | 43.87 |
> |       | 1.0  | 40.93 |
>
> ---
> **Q6**: Probability $P$: the probability is not reported nor ablated.
>
> **A6**:  We appreciate the reviewers’ request to analyze the probability $P$ used in the Standard Idempotent Module. $P$ is the probability that determines whether the second input is set to the empty signal or ground-truth label. We **perform a sensitivity ablation on Split CIFAR-100 with 500 buffer size and also focus on the ER+ID method as in Q5**. The results are shown below. In our ablation study, the second input is set to the neutral "empty" signal input $0$ with probability $P$ and the ground-truth one-hot vector $y$ with probability $1-P$. **We find that as $P$ increases, FAA consistently improves and peaks at $P=0.9$, which reaches 44.82.** Then there is a slight drop at $P=1.0$ , where FAA is 43.26.  **Given that, we choose $P=0.9$ as a default in our method.**
> |   P |   FAA |
> |----:|------:|
> | 0.2 | 20.90 |
> | 0.3 | 22.24 |
> | 0.4 | 27.03 |
> | 0.5 | 29.19 |
> | 0.6 | 31.46 |
> | 0.7 | 37.48 |
> | 0.8 | 42.24 |
> | 0.9 | 44.82 |
> | 1.0 | 43.26 |
>
> ---
> **Q7**: Backbone architecture: only ResNet18 is tested. What will happen when using a different architecture? Again, this would demonstrate broader applicability.
>
> **A7**: Thank you for your suggestion to further access broader applicability of IDER. To further validate the effectiveness of IDER, we extended our experiments to include **a fundamentally different backbone: ViT-Small**. Vision Transformers (ViTs), such as ViT-Small, are known to struggle with small datasets due to their architectural design, which demands pretraining on large-scale datasets for optimal performance. To maintain a fair comparison, we trained ViT-Small from scratch on CIFAR-100 with a buffer size of 2000 under the same CL protocol. **Given the architectural differences, we modify it to accept  the second input in a different way:** we first applied a single linear layer to project the first-pass output logits to the same embedding dimension as the [CLS] token, and then replaced the [CLS] token with this projected vector for the second pass. **Despite the inherent challenges of ViTs on CIFAR-100 and naive modification, IDER consistently delivers substantial  performance gains over the baseline ER，reinforcing its effectiveness and broader applicability.**
>
> | Method |   FAA |   FF |
> |:-------|------:|------:|
> | ER  | 13.96 | 51.86 |
> | ER+ID  | 20.95 | 37.22 |

---

> ### Author Response · Authors · 2025-11-24
> **Response to Reviewer WbzD （5/6）**
>
> **Q8**: Limited experiments on GCIL (Table 2): it would be interesting to see the results on the other considered datasets.
>
> **A8**: Thank you for the suggestion to broaden GCIL evaluation beyond CIFAR-100. We appreciate your valuable suggestion while **we consider CIFAR-100 sufficient for demonstrating effectiveness under GCIL setting. Considering many studies [1,2,3] including the paper that originally introduced GCIL setting, all their experiments under GCIL setting have been conducted on CIFAR-100.**
>
> To further assess broader applicability, **we still extended our experiments to TinyImageNet under the GCIL setting.** The results are as shown below. TinyImageNet is more challenging than CIFAR-100 dut to higher image resolution and background complexity, which amplifies difficulty under the challenging GCIL setting. The results demonstrate this with markedly lower absolute performance for standard baselines. Despite these challenges, **IDER consistently improves baselines in this harder GCIL scenarios.** These results support IDER’s broader applicability under more challenging GCIL conditions.
>
> | Method     | uniform |
> |:-----------|--------:|
> | ER         |    2.45 |
> | ER+ID      |    4.58 |
> | CLS-ER     |    7.62 |
> | CLS-ER+ID  |    8.51 |
>
> [1]Fei Mi, Lingjing Kong, Tao Lin, Kaicheng Yu, and Boi Faltings. Generalized class incremental learning. In Proceedings of the IEEE/CVF conference on computer vision and pattern recognition workshops, pp. 240–241, 2020
>
> [2]Fahad Sarfraz, Elahe Arani, and Bahram Zonooz. Sparse coding in a dual memory system for lifelong learning. In Proceedings of the AAAI Conference on Artificial Intelligence, volume 37, pp. 9714–9722, 2023.
>
> [3]Fahad Sarfraz, Elahe Arani, and Bahram Zonooz. Semantic aware representation learning for lifelong learning. In The Thirteenth International Conference on Learning Representations, 2025.
>
>
> ---
>
> **Q9**: Furthermore, why is Tiny-ImageNet not included in Table 3? Since the results from NPCL were copied from the original work, do you assure that the experimental setting is exactly the same? Otherwise, the comparison is not meaningful.
>
> **A9**: Thank you for the thoughtful questions. First, **Tiny-ImageNet is not included in Table 3 because we intentionally aligned that with NPCL’s published calibration scope, which reports ECE only for CIFAR-10/100.** We acknowledge that including Tiny-ImageNet would make Table 3 more complete and provide comprehensive evalution. Thus, We extended our experiments to TinyImageNet and also compare with recent works on uncertainty calibration for class-incremental learning [1,2] mentioned in Q3.
>
> Second, regarding protocol consistency, **our experiment setting is indeed exactly the same as NPCL, because both NPCL and our method follow the protocols in [3,4],** which are widely adopted for rehearsal-based methods. To make fair comparsion, we also unify the experimental setting from recent works on uncertainty calibration for class-incremental learning [1,2] which you suggest for comparison, including batch sizes, optimizer and learning rate. The results are shown below. **Our method consistently reduces the calibration error of baselines in Tiny-ImageNet.** We hope this could provide comprehensive evalution.
>
> | Method                        | Tinyimg |       | CIFAR-100 |       | CIFAR-10 |       |
> |:----------------------:|:-------:|:-----:|:---------:|:-----:|:--------:|:-----:|
> |                        | 500     | 4000  | 500       | 2000  | 200      | 500   |
> | DER                    | 22.80   | 10.52 | 24.84     | 10.79 | 29.91    | 16.20 |
> | Continual Calibration  | 21.32   | 16.49 | 19.43     | 19.31 | 16.39    | 12.84 |
> | T-CIL                  | 14.50   | 10.30 | 15.79     | 8.67  | 22.50    | 10.51 |
> | ER                     | 67.50   | 51.37 | 64.59     | 45.64 | 45.53    | 32.69 |
> | ER+ID (Ours)          | 21.55   | 11.14 | 13.65     | 12.87 | 12.36    | 11.73 |
> | BFP                    | 9.45    | 8.25  | 11.93     | 9.28  | 9.83     | 9.40  |
> | BFP+ID (Ours)         | 7.77    | 6.35  | 8.92      | 8.29  | 9.30     | 8.63  |
>
> [1] Li, Lanpei, et al. "Calibration of continual learning models." Proceedings of the IEEE/CVF Conference on Computer Vision and Pattern Recognition. 2024.
>
> [2] Hwang, Seong-Hyeon, Minsu Kim, and Steven Euijong Whang. "T-CIL: Temperature Scaling using Adversarial Perturbation for Calibration in Class-Incremental Learning." Proceedings of the Computer Vision and Pattern Recognition Conference. 2025.
>
> [3] Pietro Buzzega, Matteo Boschini, Angelo Porrello, Davide Abati, and Simone Calderara. Dark experience for general continual learning: a strong, simple baseline. Advances in neural information processing systems, 33: 15920–15930, 2020.
>
> [4] Matteo Boschini, Lorenzo Bonicelli, Pietro Buzzega, Angelo Porrello, and Simone Calderara. Class-incremental continual learning into the extended der-verse. IEEE transactions on pattern analysis and machine intelligence, 45(5):5497–5512, 2022.

---

> ### Author Response · Authors · 2025-11-24
> **Response to Reviewer WbzD （6/6）**
>
> **Q10**: No analysis on larger task sequences (e.g., 20 tasks on Tiny-ImageNet).
>
> **A10**: We appreciate the suggestion and have conducted additional experiments with a larger task sequence (20 tasks on Tiny-ImageNet). This setting is more challenging as the task number gets larger, which intensifies distribution shifts and increases forgetting pressure. The results demonstrate that **our method yields consistent improvements over baselines.** This analysis provides empirical evidence that underscores the robustness of our approach.
> | Method      |           FAA |
> |:-----------|-------------:|
> | ICARL      | 22.77±0.25   |
> | SCoMMER    | 32.69±0.35   |
> | SARL       | 33.23±0.98   |
> | BFP        | 39.86±0.67   |
> | XDER       | 41.75±0.38   |
> | ER         | 22.39±0.09   |
> | ER+ID      | 34.86±0.67   |
> | CLS-ER     | 41.06±0.23   |
> | CLS-ER+ID  | 41.82±0.54   |
>
>
> ---
>
> **Q11**: Figures 3, 4, and 5 are difficult to parse.
>
> **A11**: Thank you for highlighting the readability issues. In the revised version, we have **updated the captions to improve clarity and readability**. We hope these changes could make the figures easier to parse.
>
> ---
>
> **Q12**: Writing style can be overall improved.
>
> **A12**: Thank you for pointing this out. We appreciate this feedback and **revised it in detail**. In the revised version, we improved the overall writing quality by **clarifying motivation and methods, providing the detailed discussion, adding a concrete example to illustrate idempotence's effect on predictions and improving experiments and ablation studies**. We hope these changes could make the paper easier to follow.

---

> ### Comment · Reviewer_WbzD · 2025-11-25
> **Final rating**
>
> Thanks for the authors' reply. My comments have been considered and my concerns addressed. I have also gone through the revised version of the paper, the other reviewers' comments, and the authors' replies. I believe that with the new experiments, clarifications, and explanations the paper is in a much better shape than before and can be a valuable contribution to the research community. For this reason, I decided to raise my score.

---

> > ### Author Response · Authors · 2025-11-25
> > **Thank you for the review**
> >
> > Thank you very much for taking the time to re-evaluate our paper after considering our rebuttal. We greatly appreciate your positive remarks regarding the novel idea we introduced and our contribution to continual learning. We are also grateful for the improved score you've given our work. Your constructive feedback and acknowledgement of our efforts is truly encouraging.

---

### Official Review · Reviewer_aMu3 · 2025-10-30

**Soundness:** 3
**Presentation:** 3
**Contribution:** 3
**Rating:** 6
**Confidence:** 4

**Summary:**

The paper introduces IDER (Idempotent Experience Replay), a novel approach for continual learning that addresses the issue of catastrophic forgetting in replay-based methods. The key idea is to make the replay process idempotent, meaning that repeatedly revisiting the same experiences does not alter the model representation undesirably. The method achieves this by enforcing idempotent updates through the standard idempotent module and the idempotent distillation module. Experiments on several continual learning benchmarks demonstrate consistent performance gains over standard replay-based baselines.

**Strengths:**

The idea of using idempotent property to mitigate issues of poor calibration and recency bias is straightforward and intuitive.

The paper is well-written and easy to follow.

The proposed method is shown to be effective.

**Weaknesses:**

As stated in Lines 71–74, the paper claims a strong correlation between the idempotence distance and prediction error. However, it remains unclear whether this relationship has been formally analysed. Could the authors provide empirical evidence or theoretical justification for this claim?

To enable idempotence with respect to the second input, the proposed method divides the backbone into two parts. What principles guide this division? Does choosing different partition points (e.g., splitting at shallower or deeper layers) affect model performance or stability? It would be helpful if the authors could provide theoretical insight or experimental analysis on how the division point influences the effectiveness of idempotent learning.

Lines 207–209 describe the probability P that determines whether the second input is set to the ground-truth label or the empty signal. Is there any hyperparameter analysis that explores the sensitivity of the method to P? How should this value be selected in practice?

The overall loss function combines three components: the Standard Idempotent Module, Idempotent Distillation Module, and Experience Replay. Are there ablation experiments isolating the contribution of each component? Furthermore, the paper states that the proposed method primarily addresses poor calibration and recency bias. However, these issues are often mitigated in modern replay-based methods. Why does this approach not integrate or compare directly with more recent baselines such as L2P [1], HIDE-prompt [2], or VQ-prompt [3]? Has the proposed method been evaluated under online continual learning settings to validate its general applicability?

Table 1 shows improvements when integrating IDER with ER, BFP, and CLS-ER. However, could the proposed method also be compatible with other state-of-the-art baselines such as SCoMMER or SARL? It would be valuable to verify whether the idempotent mechanism consistently enhances these methods as well.

Regarding the experiments, I noted that the evaluation is primarily conducted on small-scale datasets like the CIFAR series and Tiny-ImageNet. I encourage the authors to validate the method's effectiveness on larger-scale and more diverse datasets. Furthermore, a concrete example illustrating how the proposed method achieves error correction in practice would greatly enhance the paper’s clarity and impact.

[1] Wang et al., Learning to prompt for continual learning. CVPR 2022

[2] Wang et al., Hierarchical Decomposition of Prompt-Based Continual Learning: Rethinking Obscured Sub-optimality. NIPS 2023

[3] Li et al., Vector Quantization Prompting for Continual Learning. NIPS 2024

**Questions:**

Please refer to the weakness section.

---

> ### Author Response · Authors · 2025-11-24
> **Response to Reviewer aMu3 （1/4）**
>
> Thank you for review and insightful questions. We would like to respond your questions point by point below.
>
> ---
> **Q1**: As stated in Lines 71–74, the paper claims a strong correlation between the idempotence distance and prediction error. However, it remains unclear whether this relationship has been formally analysed. Could the authors provide empirical evidence or theoretical justification for this claim?
>
> **A1**：Thank you for your question. The statement in Lines 71–74—that the idempotence distance correlates strongly with prediction error—**originates from prior work [1], which is supported empirically.** Specifically, as Figure 15 in [1] shows,  **the Pearson correlation between idempotence error and accuracy is −0.94 on ImageNet-C, indicating a strong negative trend.** We have already cited [1] in our paper when stating this claim, and we have revised the phrasing to make it more clear in the revised version.
>
>
> [1] Nikita Durasov, Doruk Oner, Jonathan Donier, Hieu Le, and Pascal Fua. Enabling uncertainty estimation in
> iterative neural networks. In Forty-first International Conference on Machine Learning, 2024
>
>
> ---
>
> **Q2**： To enable idempotence with respect to the second input, the proposed method divides the backbone into two parts. What principles guide this division? Does choosing different partition points (e.g., splitting at shallower or deeper layers) affect model performance or stability?  It would be helpful if the authors could provide theoretical insight or experimental analysis on how the division point influences the effectiveness of idempotent learning.
>
> **A2**: Thank you for your insightful questions.
>
> First, we divide the backbone at a **mid-layer based on the assumption that shallower splits amplify noise and destabilize training while deeper splits attenuate the second signal and weaken idempotence effect.**
>
> Second, to support this assumption empirically, we conduct ablation studies for different partition points across different datasets. The results are shown below. **The results empirically demonstrate that mid-layer partitions achieve the best trade-off, yielding consistently best performance in Final Average Accuracy(FAA) across different datasets while choosing shallower or deeper partition points would reduce the FAA and affect model performance.**
>
> Overall, we hope that this ablation study could provide help for dividing the backbone.
>
> | Method               | CIFAR-100 | Tiny-ImageNet |
> |:---------------------|----------:|-------------:|
> | shallower(5th layer) |    40.51 |        42.10 |
> | deeper(13th layer)   |    43.09 |        42.28 |
> | Ours (9th layer)     |    44.82 |        43.05 |
>
>
> ---
>
> **Q3**: Lines 207–209 describe the probability P that determines whether the second input is set to the ground-truth label or the empty signal. Is there any hyperparameter analysis that explores the sensitivity of the method to P? How should this value be selected in practice?
>
> **A3**: Thanks for thoughtful question. **Yes.** P is the probability that determines whether the second input is set to the empty signal or ground-truth label. We appreciate the reviewers’ request to analyze the probability P used in the Standard Idempotent Module.  We perform a sensitivity ablation on Split CIFAR-100 with 500 buffer size and focus on the ER+ID method. The results are shown below. In our ablation study, **the second input is set to the neutral "empty" signal input 0 with probability $P$ and the ground-truth one-hot vector y with probability $1-P$.** We find that as P increases, FAA consistently improves and peaks at P=0.9, which reaches 44.82. Then there is a slight drop at P=1.0 , where FAA is 43.26. **Practically, we recommend $P∈[0.8, 1.0]$, with P=0.9 as a robust default that balances training stability with sufficient variability.**
> | P   |   FAA |
> |:----|------:|
> | 0.2 | 20.90 |
> | 0.3 | 22.24 |
> | 0.4 | 27.03 |
> | 0.5 | 29.19 |
> | 0.6 | 31.46 |
> | 0.7 | 37.48 |
> | 0.8 | 42.24 |
> | 0.9 | 44.82 |
> | 1.0 | 43.26 |

---

> ### Author Response · Authors · 2025-11-24
> **Response to Reviewer aMu3 （2/4）**
>
> **Q4**: The overall loss function combines three components: the Standard Idempotent Module, Idempotent Distillation Module, and Experience Replay. Are there ablation experiments isolating the contribution of each component?
>
> **A4**: Thank you for your suggestion. We conduct a component ablation study to isolate the contributions of each part of the overall objective: the Standard Idempotent Module (SIM), the Idempotent Distillation Module (IDM), and Experience Replay (ER). We focus on the ER+ID method for this study, as it does not introduce any additional components. The ablation study results are as follows.
>
> First, using only the Standard Idempotent Module (SIM) or combing SIM with ER produces similar performance compared with finetuning or ER baseline, indicating that **SIM alone trains model to be idempotent on the current task, which making it well-suited for subsequent idempotent distillation and  doesn't mitigate catastrophic forgetting.** It also demonstrates that **modified architecture does not influence performance and the observed improvements benefit from idempotent distillation loss.**
>
> Second, adding the Idempotent Distillation Module (IDM) yields substantial performance gains, which further improves accuracy to 44.82%. **This validates the effectiveness of idempotence distillation loss, as it is not only designed for enforcing idempotence in CL and can serve as a distillation loss but also prevents training collapse and bias error amplification.** More details are provided in appendix F.
>
> | SIM | IDM | ER  |   FAA |
> |:---:|:---:|:---:|------:|
> |  ✓  |  ×  |  ×  |  8.23 |
> |  ✓  |  ×  |  ✓  | 24.73 |
> |  ✓  |  ✓  |  ✓  | 44.82 |
>
>
> ---
>
> **Q5**: Furthermore, the paper states that the proposed method primarily addresses poor calibration and recency bias. However, these issues are often mitigated in modern replay-based methods. Why does this approach not integrate or compare directly with more recent baselines such as L2P [1], HIDE-prompt [2], or VQ-prompt [3]?
>
> **A5**: Thank you for your question. Our method is designed to operate in the challenging scenarios where the model starts from a **randomly initialized state** while the modern replay-based methods mentioned are all based on **pretrained model and expanding architecutre while keeping most of the pretrained backbone frozen to learn new task knowledge, which provide a substantial advantage in CIL settings.** These pre-trained backbones often include representations of a significant portion of the classes encountered later in CL tasks or for classes that are highly similar. As a result, these issues such as poor calibration and recency bias are largely bypassed in such approaches.
>
> [1] Wang et al., Learning to prompt for continual learning. CVPR 2022
>
> [2] Wang et al., Hierarchical Decomposition of Prompt-Based Continual Learning: Rethinking Obscured Sub-optimality. NIPS 2023
>
> [3] Li et al., Vector Quantization Prompting for Continual Learning. NIPS 2024
>
> ---
>
> **Q6**: Has the proposed method been evaluated under online continual learning settings to validate its general applicability?
>
> **A6**: Thank you for your suggestion. Unlike traditional class incremental learning (CIL), which typically allows multiple epochs per task and revisiting buffered samples, online continual learning (Online CL) enforces a single-pass data stream. While our method focuses on batch training on sequential taks, to validate its general applicability, we conduct experiments on CIFAR-100 under online continual learning following SARL. The results are shown below. **Our method consistently improves both ER and SARL baselines with different batch sizes.** The consistent gains indicate robustness and practical applicability of our method in realistic continual learning scenarios.
>
> | Method    |      Buffer    1000 |      Buffer    2000 |
> |:----------|---------------:|---------------:|
> | ER        | 16.07±0.88    | 18.85±0.27    |
> | ER+ID     | 17.59±0.91    | 19.51±0.45    |
> | SARL      | 24.39±1.44    | 26.39±1.03    |
> | SARL+ID   | 24.87±0.73    | 26.72±1.16    |

---

> ### Author Response · Authors · 2025-11-24
> **Response to Reviewer aMu3 （3/4）**
>
> **Q7** : Table 1 shows improvements when integrating IDER with ER, BFP, and CLS-ER. However, could the proposed method also be compatible with other state-of-the-art baselines such as SCoMMER or SARL? It would be valuable to verify whether the idempotent mechanism consistently enhances these methods as well.
>
> **A7**: Thank you for pointing it out. **Yes, our method is indeed compatible with other state-of-the-art baselines and we have already intergrated our method with SARL under generalized class incremental learning (GCIL) setting in Table 2.** To further verify this, we conduct the experiment to compare SARL and SARL intergrated with our method across different datasets and buffer sizes under class incremental learning (CIL) setting. **The results show consistent improvements for SARL when our method is integrated.**  Although SARL already employs several additional loss terms to boost performance leading to the modest gains, our method still delivers consistent benefits, which demonstrate the robustness and effectiveness of our method.
>
> | Dataset    | Buffer Size | SARL          | SARL+ID       |
> |------------|-------------|---------------|---------------|
> | CIFAR-10   | 200         | 68.87±1.37    | 69.20±0.85    |
> | CIFAR-10   | 500         | 73.98±0.46    | 74.35±0.42    |
> | CIFAR-100  | 500         | 46.69±0.79    | 47.23±0.53    |
> | CIFAR-100  | 2000        | 57.06±0.48    | 57.56±0.67    |
> | TinyImg    | 500         | 28.44±2.30    | 28.92±0.96    |
> | TinyImg    | 4000        | 38.83±0.81    | 40.69±0.73    |
>
>
> ---
>
> **Q8**: Regarding the experiments, I noted that the evaluation is primarily conducted on small-scale datasets like the CIFAR series and Tiny-ImageNet. I encourage the authors to validate the method's effectiveness on larger-scale and more diverse datasets.
>
> **A8**: Thank you for valuable suggestion. We agree that incorporating larger-scale datasets can validate the effectiveness of our method. However, **we consider Split Tiny-ImageNet to be a substantial dataset. Numerous recent studies [1,2,3] have utilized Split Tiny-ImageNet as the largest dataset in their experiments.**
>
> To further validate our method's effectiveness, **we conduct additional experiments on Split ImageNet-R.** ImageNet-R has larger resolution and intentionally includes out-of-distribution (OOD) samples, making it a more challenging and realistic benchmark. Recent works [4] also adopt Split ImageNet-R as the largest and most challenging dataset in their experiments. **Despite the ResNet-18 backbone struggling with ImageNet-R, integrating our method into the ER baseline yields significant improvements,** demonstrating strong robustness and effectiveness of our method in difficult settings.
>
> | Method |   FAA |    FF |
> |:-------|------:|------:|
> | ER     |  5.90 | 38.05 |
> | ER+ID  | 13.86 | 31.97 |
>
> [1] Liu, Ruiqi, et al. "Continual learning in the frequency domain." Advances in Neural Information Processing Systems 37 (2024): 85389-85411.
>
> [2]Fahad Sarfraz, Elahe Arani, and Bahram Zonooz. Semantic aware representation learning for lifelong learning. In The Thirteenth International Conference on Learning Representations, 2025.
>
> [3]Hwang, Seong-Hyeon, Minsu Kim, and Steven Euijong Whang. "T-CIL: Temperature Scaling using Adversarial Perturbation for Calibration in Class-Incremental Learning." Proceedings of the Computer Vision and Pattern Recognition Conference. 2025.
>
> [4] Kang, Hankyul, et al. "Do Your Best and Get Enough Rest for Continual Learning." Proceedings of the Computer Vision and Pattern Recognition Conference. 2025.

---

> > ### Author Response · Authors · 2025-11-24
> > **Response to Reviewer aMu3 （4/4）**
> >
> > **Q9**: Furthermore, a concrete example illustrating how the proposed method achieves error correction in practice would greatly enhance the paper’s clarity and impact.
> >
> > **A9**: Thank you for the helpful suggestion. To concretely illustrate error correction, we compare ER and our ER+ID on Split-CIFAR100 with 10 incremental tasks (task t contains classes ids from (t−1)·10 to t·10). In this setup, the first 10 classes (class ids from 0 to 9) belong to task 1 and the next 10 classes (class ids from 10 to 19) belong to task 2. **We then evaluate a test example from class id=4 (an old class from task 1).  The top-5 predictions from ER and ER+ID are reported below and Visualization is provided in revised paper Figure 8.**
> >
> > ### Top-5 Predictions Comparison
> >
> > ### ER Baseline
> > | Rank | Class | Logit   |
> > |------|-------|---------|
> > | 1    | 18    | 15.3692 |
> > | 2    | 14    | 11.9968 |
> > | 3    | 12    | 11.1726 |
> > | 4    | 19    | 10.7846 |
> > | 5    | 17    | 10.2170 |
> >
> > ### ER+ID
> > | Rank | Class | Logit   |
> > |------|-------|---------|
> > | 1    | 4     | 12.3276 |
> > | 2    | 3     | 9.5518  |
> > | 3    | 18    | 9.2582  |
> > | 4    | 8     | 7.1098  |
> > | 5    | 6     | 6.8284  |
> >
> > **The top-5 predictions show ER’s strong recency bias toward new classes: Rank1–5 = {18: 15.3692, 14: 11.9968, 12: 11.1726, 19: 10.7846, 17: 10.2170}, misclassifying the sample.** In contrast, ER+ID corrects the error and calibrates confidence: Rank1–5 = {4: 12.3276, 3: 9.5518, 18: 9.2582, 8: 7.1098, 6: 6.8284}, **with the true class promoted to top-1 prediction and overconfidence on new classes notably reduced.**

---

> > > ### Comment · Reviewer_aMu3 · 2025-11-28
> > > **Thank you for the response**
> > >
> > > I would like to thank the authors for the response. These answer my questions. I hope to see this reflected in the final version. I am happy to increase my score.

---

> > > > ### Author Response · Authors · 2025-11-28
> > > > **Thank for the review**
> > > >
> > > > Thank you very much for the constructive comments and insightful questions. We sincerely appreciate the time and effort you've dedicated to this. We will ensure that the new results and analysis are incorporated into the updated version.

---

### Official Review · Reviewer_xwPR · 2025-10-31

**Soundness:** 2
**Presentation:** 3
**Contribution:** 3
**Rating:** 6
**Confidence:** 4

**Summary:**

This manuscript introduces the idempotent property to mitigate catastrophic forgetting, a fundamental challenge in incremental learning. The authors design two complementary loss functions: one enforces idempotence on current data, while the other distills this property from a previous model checkpoint using a memory buffer of historical samples. Extensive experiments are conducted to validate the effectiveness of the proposed approach.

**Strengths:**

• The manuscript introduce idempotence, a mathematical property, to tackle catastrophic forgetting and poor model calibration in continual learning.
• The proposed IDER is a lightweight framework and functions as a plug-and-play module for performance gains

**Weaknesses:**

• The paper primarily relies on intuition and empirical success to introduce idempotence. It lacks a rigorous theoretical analysis or hypothesis for why enforcing output stability should directly mitigate catastrophic forgetting at a fundamental level.
• The empirical validation is comprehensive on CIFAR-10, CIFAR-100, and Tiny-ImageNet. However, to firmly establish the method's practicality and generalizability, evaluation on a large-scale dataset, e.g., ImageNet-1K.
• The hyperparameter sensitivity analysis for α and β is relatively brief.
• The experimental comparisons are heavily focused on replay-based methods, which is natural as IDER is a plug-in for this paradigm. However, comparing against strong representatives from other CL fmethods, such as regularization-based methods or memory-free approaches, would more comprehensively position IDER's contribution within the entire field and highlight its unique value.

**Questions:**

• A simple theoretical proposition or a more in-depth discussion connecting the idempotence loss to established continual learning theory would significantly strengthen the foundation.
• Test IDER in settings like online continual learning or with tasks containing out-of-distribution samples to better probe its limits and robustness.
• Perform a more systematic hyperparameter sensitivity analysis, showing how performance varies with different values of α and β across key datasets.
• Add comparisons with regularization-based and memory-free methods in the main experiments.

**Details Of Ethics Concerns:**

I did not identify any ethical concerns related to this paper. The work does not involve sensitive data, human subjects, or
potentially harmful application.

---

> ### Author Response · Authors · 2025-11-24
> **Response to Reviewer xwPR （1/4）**
>
> Thank you for review and valuable suggestions. We would like to respond to your questions below.
>
>
> ---
> **Q1**: The paper primarily relies on intuition and empirical success to introduce idempotence. It lacks a rigorous theoretical analysis or hypothesis for why enforcing output stability should directly mitigate catastrophic forgetting at a fundamental level.
>
> **A1**: Thank you for your valuable comments. Below, we provide detailed explanations on why idempotence can help continual learning resist catastrophic forgetting and improve uncertainty calibration. More details can be seen in appendix F.
>
> In CL, models are often poorly calibrated and over-confident, a problem exacerbated by recency bias toward new tasks. To mitigate this issue, we require **the model to maintain stable predictions on data from previous tasks even after parameter updates induced by new knowledge, as self-consistency indicates that the network's output is aligned with the learned in-distribution manifold and can make reliable (well-calibrated) prediction. This condition can be translated into enforcing idempotence.** We can formalize the desired idempotence condition as $f_t(x, f_t(x, 0)) = f_t(x, 0)$, where $x$ is from both previous and current tasks and $f_t$ represents the current model. In practice, as data from previous tasks can't be obtained, the loss can be defined as:
>
> $$\mathcal{L} = \sum_{(x,y)\in \mathcal{T}_{t,M}} \left\lVert f_t(x, 0) - f_t(x, f_t(x, 0)) \right\rVert_2^2,$$
>
> where $M$ is the buffer memory which stores data from previous tasks.
>
> Minimizing this loss drives the network toward the condition which is needed for model to make reliable predictions in CL. However, minimizing the idempotence loss in CL is not trivial. First, we propose $\mathcal{L}_{\text{ice}}$ to train model idempotent for sequential tasks. Second, we modify the idempotence distillation loss by using the model checkpoint at the end of the last task for the second application, which can be rewritten as
>
> $$\mathcal{L}&#95;{\text{ide}} = \sum&#95;{(x,y)\in \mathcal{T}&#95;{t,M}} \left\lVert f&#95;t(x, 0) - f&#95;{t-1}(x, f&#95;t(x, 0)) \right\rVert&#95;2^2.$$
>
> The modification has two benefits:
>
> **First, it prevents training collapse and bias error amplification.** Consistent with [1] and [2], directly optimizing the idempotence loss induces two gradient pathways: (1) A desirable pathway that updates $f_{t}(x, 0)$ toward the correct in-distribution manifold. (2) An undesirable pathway that may cause the manifold to expand, thereby including an incorrect $f_{t}(x, 0)$.  **To counteract the latter gradient pathways, a frozen copy of the network is often used.**
>
> **Second, it is designed for enforcing idempotence in CL and can serve as a distillation loss.** According to $\mathcal{L}_{\text{ice}}$, under empirical risk minimization, we can assume that:
>
>   $$f_t(x, f_t(x, \mathbf{0})) = f_t(x, \mathbf{0}).$$
>
>   Thus, we rewrite the $\mathcal{L}_{\text{ide}}$ as:
>
> $$\mathcal{L}&#95;{\text{ide}} = \sum&#95;{(x,y)\in \mathcal{T}&#95;{t,M}} \left\lVert f&#95;t(x,f&#95;t(x, 0)) - f&#95;{t-1}(x, f&#95;t(x, 0)) \right\rVert&#95;2^2.$$
>
>   **First, according to the same input for $f_t$ and $f_{t-1}$ in this equation, this idempotent distillation loss could serve as a standard regularization loss**:
>
> $$\mathcal{L}&#95;{\text{re}} = \sum&#95;{(x,y)\in \mathcal{T}&#95;{t,M}} \left\lVert f&#95;t(x) - f&#95;{t-1}(x) \right\rVert&#95;2^2,$$
>
>   **which is often used in CL methods[3,4] to mitigate catastrophic forgetting**. Second, when incorporating the second input that conveys logits from $f_t$, the loss steers the current model $f_t$ to update in a direction where the predictions remain correctly interpretable by the previous model $f_{t-1}$. **Consequently, in sequential tasks, $f_{t-1}$ and $f_t$ are driven toward idempotence, feeding back $f_t$'s own output does not alter the prediction of $f_{t-1}$, yielding more reliable predictions on data from previous tasks to improve calibration in continual learning**.
>
> [1] Shocher, Assaf, et al. "Idempotent generative network." In The Thirteenth International Conference on Learning Representations, 2024.
>
> [2] Durasov, Nikita, et al. "IT³: Idempotent Test-Time Training." ICML 2025.
>
> [3] Gu, Qiao, Dongsub Shim, and Florian Shkurti. "Preserving linear separability in continual learning by backward feature projection." Proceedings of the IEEE/CVF Conference on Computer Vision and Pattern Recognition. 2023.
>
> [4] Sarfraz, Fahad, Elahe Arani, and Bahram Zonooz. "Semantic Aware Representation Learning for Lifelong Learning." The Thirteenth International Conference on Learning Representations.

---

> > ### Author Response · Authors · 2025-11-24
> > **Response to Reviewer xwPR （2/4）**
> >
> > **Q2**: The empirical validation is comprehensive on CIFAR-10, CIFAR-100, and Tiny-ImageNet. However, to firmly establish the method's practicality and generalizability, evaluation on a large-scale dataset, e.g., ImageNet-1K.
> >
> > **A2**: Thank you for valuable suggestion. We agree that incorporating larger-scale datasets can validate the generalizability of our method. However, **we consider Split Tiny-ImageNet to be a substantial dataset. Numerous recent studies [1,2,3] have utilized Split Tiny-ImageNet as the largest dataset in their experiments.**
> >
> > To further validate our method's effectiveness, **we conducted additional experiments on Split ImageNet-R. ImageNet-R has larger resolution and intentionally includes out-of-distribution (OOD) samples**, making it a more challenging and realistic benchmark. **And this also satisfies the insightful suggestion you provide in Q5: Test IDER in settings with tasks containing out-of-distribution samples.** Recent works [4] also adopt Split ImageNet-R as the largest and most challenging dataset in their experiments. **Despite the ResNet-18 backbone struggling with ImageNet-R, integrating our method into the ER baseline yields significant improvements**, demonstrating strong robustness and effectiveness of our method in difficult settings.
> >
> > | Method |   FAA |    FF |
> > |:-------|------:|------:|
> > | ER     |  5.90 | 38.05 |
> > | ER+ID  | 13.86 | 31.97 |
> >
> > [1] Liu, Ruiqi, et al. "Continual learning in the frequency domain." Advances in Neural Information Processing Systems 37 (2024): 85389-85411.
> >
> > [2]Fahad Sarfraz, Elahe Arani, and Bahram Zonooz. Semantic aware representation learning for lifelong learning. In The Thirteenth International Conference on Learning Representations, 2025.
> >
> > [3]Hwang, Seong-Hyeon, Minsu Kim, and Steven Euijong Whang. "T-CIL: Temperature Scaling using Adversarial Perturbation for Calibration in Class-Incremental Learning." Proceedings of the Computer Vision and Pattern Recognition Conference. 2025.
> >
> > [4] Kang, Hankyul, et al. "Do Your Best and Get Enough Rest for Continual Learning." Proceedings of the Computer Vision and Pattern Recognition Conference. 2025.
> >
> >
> > ---
> >
> > **Q3**: The hyperparameter sensitivity analysis for α and β is relatively brief.
> >
> > **A3**: Thank you for highlighting the need for a more thorough $\alpha$ and $\beta$ hyperparameter sensitivity analysis and we now have conducted ablation study on the $\alpha$ and $\beta$ introduced in Equation 8. **We focus on the ER+ID method for this study, as it  consistently yields substantial improvements over ER baseline across all datasets and does not introduce any additional hyperparameters.** We conduct tests on the Split CIFAR-100 with a buffer size of 500. The ablation study results are shown below. **Our choice of hyperparameters in the paper yields optimal performance, and our methods demonstrates robustness to these hyperparameter selections.** We hope that these ablation studies will provide the reviewer with a more comprehensive understanding of our work.
> >
> > | alpha | beta | FAA   |
> > |-------|------|-------|
> > | 0.1   | 0.1  | 44.77 |
> > |       | 0.2  | 44.41 |
> > |       | 0.5  | 43.82 |
> > |       | 1.0  | 41.20 |
> > | 0.2   | 0.1  | 44.24 |
> > |       | 0.2  | 44.43 |
> > |       | 0.5  | 44.33 |
> > |       | 1.0  | 41.58 |
> > | 0.5   | 0.1  | 41.74 |
> > |       | 0.2  | 44.25 |
> > |       | 0.5  | 44.82 |
> > |       | 1.0  | 42.72 |
> > | 1.0   | 0.1  | 40.42 |
> > |       | 0.2  | 43.02 |
> > |       | 0.5  | 43.87 |
> > |       | 1.0  | 40.93 |

---

> > > ### Author Response · Authors · 2025-11-24
> > > **Response to Reviewer xwPR （3/4）**
> > >
> > > **Q4**: The experimental comparisons are heavily focused on replay-based methods, which is natural as IDER is a plug-in for this paradigm. However, comparing against strong representatives from other CL methods, such as regularization-based methods or memory-free approaches, would more comprehensively position IDER's contribution within the entire field and highlight its unique value.
> > >
> > > **A4**: Thank you for your valuable suggestion regarding the comparison with a broader range of continual learning (CL) methods, including regularization-based or memory-free approaches. We appreciate your insights, as they help in assessing the robustness of our proposed method.
> > >
> > > First,  in the current CL taxonomy, **memory-free approaches typically includes both regularization-based methods which utilize distillation or stability constraints without replay and parameter-isolation approaches based on architecture expansion.** As a result, **we understand your suggestion as asking for comparisons against regularization-based and parameter-isolation methods.**
> > >
> > > Second, it is important to clarify that **comparing parameter-isolation approaches with rehearsal-based methods is not fair.** Parameter-isolation approaches often depend on **architecture expansion** to learn new knowledge while rehearsal-based methods usually retain a fixed architecture, making direct comparisons sensitive to these structural differences. In addition, most recent parameter-isolation approaches[1] [2] depend on **pretrained backbones**. These pre-trained backbones often include representations of a significant portion of the classes encountered later in CL tasks or classes that are highly similar, which provides prior advantages unrelated to the CL mechanism itself. **As a result, in comparison to rehearsal-based methods, the parameter-isolation methods have significant advantages.** What's more, **parameter-isolation methods are evaluated under different experimental settings, further undermining direct comparisons.**
> > >
> > > Third, we **compare our method against some state-of-art regularization-based methods under the same experiment setting as ours**: LwF+NCM[3], LwF+SDC[4], PASS[5], FeTrIL[6], FeCAM[7], EFC[8], LwF+LDC[9]. For our methods, the buffer size is set to 2000 on CIFAR-100 and 4000 on Tiny-ImageNet. **The results below show that our methods generally achieve better performance.**
> > >
> > > | Method      | CIFAR100       | Tiny-ImageNet  |
> > > |:-----------:|:--------------:|:--------------:|
> > > | LwF+NCM     | 40.5±2.7       | 28.6±1.1       |
> > > | LwF+SDC     | 40.6±1.8       | 29.5±0.8       |
> > > | PASS        | 37.8±0.2       | 31.2±0.4       |
> > > | FeTrIL      | 37.0±0.6       | 24.4±0.6       |
> > > | FeCAM       | 33.1±0.9       | 24.9±0.5       |
> > > | EFC         | 43.6±0.7       | 34.1±0.8       |
> > > | LwF+LDC     | 45.4 ± 2.8       | 34.2±0.7       |
> > > | ER+ID       | 56.59±0.35     | 43.05±1.40     |
> > > | BFP+ID      | 57.74±0.64     | 43.51±0.59     |
> > > | CLS-ER+ID   | 56.36±0.78     | 46.17±0.22     |
> > >
> > > [1] Liang, Yan-Shuo, and Wu-Jun Li. "Inflora: Interference-free low-rank adaptation for continual learning." Proceedings of the IEEE/CVF Conference on Computer Vision and Pattern Recognition. 2024.
> > >
> > > [2]Zhou, Da-Wei, et al. "Expandable subspace ensemble for pre-trained model-based class-incremental learning." Proceedings of the IEEE/CVF Conference on Computer Vision and Pattern Recognition. 2024.
> > >
> > > [3]Rebuffi, S.A., Kolesnikov, A., Sperl, G., Lampert, C.H.: icarl: Incremental classifier and representation learning. In: Conference on Computer Vision and Pattern Recognition
> > >
> > > [4]Yu, L., Twardowski, Bartlomiej, Liu, Xialei, Herranz, L., Wang, K., Jui, S., Weijer, J.v.d.: Semantic drift compensation for class-incremental learning. In: Proceedings of the IEEE/CVF Conference on Computer Vision and Pattern Recognition. pp. 6982–6991 (2020)
> > >
> > > [5]Zhu, F., Zhang, X.Y., Wang, C., Yin, F., Liu, C.L.: Prototype augmentation and self-supervision for incremental learning. In: Proceedings of the IEEE/CVF Conference on Computer Vision and Pattern Recognition. pp. 5871–5880 (2021)
> > >
> > > [6]Petit, G., Popescu, A., Schindler, H., Picard, D., Delezoide, B.: Fetril: Feature translation for exemplar-free class-incremental learning. In: Proceedings of the IEEE/CVF Winter Conference on Applications of Computer Vision. pp. 3911– 3920 (2023)
> > >
> > > [7]Goswami, D., Liu, Y., Twardowski, B., van de Weijer, J.: Fecam: Exploiting the heterogeneity of class distributions in exemplar-free continual learning. Advances in Neural Information Processing Systems 36 (2024)
> > >
> > > [8]Magistri, S., Trinci, T., Soutif-Cormerais, A., van de Weijer, J., Bagdanov, A.D.: Elastic feature consolidation for cold start exemplar-free incremental learning. In: International Conference on Learning Representations (2024)
> > >
> > > [9]Gomez-Villa, Alex, et al. "Exemplar-free continual representation learning via learnable drift compensation." European Conference on Computer Vision. Cham: Springer Nature Switzerland, 2024.

---

> > > > ### Author Response · Authors · 2025-11-24
> > > > **Response to Reviewer xwPR （4/4）**
> > > >
> > > > **Q5**: Test IDER in settings like online continual learning or with tasks containing out-of-distribution samples to better probe its limits and robustness.
> > > >
> > > > **A5**: Thank you for your valuable suggestions.
> > > >
> > > > First,  **we test IDER under online continual learning setting**. Unlike traditional class incremental learning (CIL), which typically allows multiple epochs per task and revisiting buffered samples, online continual learning (Online CL) enforces a single-pass data stream. While our method focuses on batch training on sequential taks, to validate its general applicability, we conduct experiments on CIFAR-100 under online continual learning following SARL. The results are shown below. **Our method consistently improves both ER and SARL baselines with different batch sizes.**  The consistent gains indicate robustness and practical applicability of our method in realistic continual learning scenarios.
> > > >
> > > > | Method    |      Buffer    1000 |      Buffer    2000 |
> > > > |:----------|---------------:|---------------:|
> > > > | ER        | 16.07±0.88    | 18.85±0.27    |
> > > > | ER+ID     | 17.59±0.91    | 19.51±0.45    |
> > > > | SARL      | 24.39±1.44    | 26.39±1.03    |
> > > > | SARL+ID   | 24.87±0.73    | 26.72±1.16    |
> > > >
> > > > Second, to test IDER in settings with tasks containing out-of-distribution samples, **we conduct experiments on Split ImageNet-R.** **ImageNet-R has larger resolution and intentionally includes out-of-distribution (OOD) samples**, making it a more challenging and realistic benchmark. **Despite the ResNet-18 backbone struggling with ImageNet-R, integrating our method into the ER baseline yields significant improvements**, demonstrating strong robustness and effectiveness of our method in difficult settings.
> > > >
> > > > | Method |   FAA |    FF |
> > > > |:-------|------:|------:|
> > > > | ER     |  5.90 | 38.05 |
> > > > | ER+ID  | 13.86 | 31.97 |

---

> ### Comment · Reviewer_xwPR · 2025-11-28
> **Raise my score**
>
> I would like to thank the authors for their thoughtful responses to my comments. All of my concerns have been carefully considered and satisfactorily addressed. This work presents a clear advancement in the field, and I believe it will be of great value to the research community. Therefore, I am pleased to raise my score.

---

> > ### Author Response · Authors · 2025-11-28
> > **Thanks for raising your score**
> >
> > We are very glad to hear that all of your concerns have been addressed. Thank you for your thoughtful review and constructive suggestions, which have greatly improved the quality and clarity of our manuscript. We sincerely appreciate the time and effort you've dedicated to this. Thanks again for your review and comments.

---

### Official Review · Reviewer_f4Fu · 2025-11-01

**Soundness:** 3
**Presentation:** 3
**Contribution:** 4
**Rating:** 10
**Confidence:** 3

**Summary:**

This paper aims to improve accuracy and calibration in continual learning by extending idempotence enforcing loss function into the continual learning setting with experience replay. A function is called idempotent if consecutive applications of it gives the same result as one application.
In the single task setting, the idempotence enforcing loss function encourages the model $f(x, y)$ to be idempotent with respect to its second input $y$. This work introduces a new loss function to adapt adaptation of this idea in the continual learning setting. This new loss function can be combined with some existing techniques to improve them
The improved calibration and accuracy claims are supported by experimental results.

**Strengths:**

The paper is well written and organized. Adaptation of the idempotence loss function to the continual learning setting is creative and nuanced. The experimental results are promising.

**Weaknesses:**

I think giving more intuition about the loss function would be helpful for the reader. For example, in lines 250-252, it is stated that minimizing equation 5 biases $f_t$ towards the wrong label (even though with probability $1-P$ that objective would be minimized in eq 5), but why would $f_{t-1}$ not have the same problems? Expanding that explanation would be great.

**Questions:**

Please see the weakness section.
In general providing more intuition, especially on why this loss function helps with calibration would be great.

---

> ### Author Response · Authors · 2025-11-24
> **Response to Reviewer f4Fu**
>
> Thank you very much for your positive and encouraging feedback. We appreciate your recognition of the paper’s motivation and readability, and we’re glad that you found the creativity to introduce idempotence into continual learning.
>
>
> ---
>
> **Q1**: I think giving more intuition about the loss function would be helpful for the reader. For example, in lines 250-252, it is stated that minimizing equation 5 biases $f_t$ towards the wrong label (even though with probability $1-P$ that objective would be minimized in eq 5), but why would $f_{t-1}$ not have the same problems? Expanding that explanation would be great.
>
> **A1**: Thank you for valuable suggestions. We have **revised our paper in method section to give more explanations, visualizing concrete examples to illustrate how idempotence helps prediction and provide detailed discussion in appendix F**.
>
> First,  we explain the reason that causes the issue.  Consistent with [1] and [2], directly optimizing the idempotence loss induces two gradient pathways: (1) A desirable pathway that updates $f_{t}(x, 0)$ toward the correct in-distribution manifold. (2) An undesirable pathway that may cause the manifold to expand, thereby including an incorrect $f_{t}(x, 0)$. **The undersirable pathway would cause the bias error amplification.**
> Second, since **$f_{t-1}$ is the model checkpoint at the end of the last task and  is frozen**, we only optimize $f_{t}$. **Because $f_{t-1}$ is approximately idmepotent for data from previous tasks, which preserve more previous knowledge and make stable predictions for data from previous tasks.** Thus, when traning on data from memory, using it as the second pass model does not “chase” $f_{t}$ errors caused by recency bias on new classes, instead it would pull the logits back toward stable in-distribution manifold in the way it is trained at the last task. Moreover, by using $f_{t-1}$, idempotent loss in eq 6 can **serve as distillation loss that steers the current model $f_t$ to update in a direction where the predictions remain correctly interpretable by the previous model $f_{t-1}$.**
>
> [1] Shocher, Assaf, et al. "Idempotent generative network." In The Thirteenth International Conference on Learning Representations, 2024.
>
> [2] Durasov, Nikita, et al. "IT³: Idempotent Test-Time Training." ICML 2025.

---

### Author Response · Authors · 2025-12-02
**Rebuttal Summary for Area Chair**

Dear AC and all reviewers,

We wish to express our gratitude for overseeing the review process of our submission and deeply appreciate the time and effort invested by you in assessing our work. We also sincerely appreciate the constructive comments and insightful questions from reviewers, which substantially improved the quality and clarity of our manuscript.


Our paper received **initial scores as 10, 6 , 6, 2.** All reviewers provided valuable and insightful comments. Based on these suggestions, we prepared a detailed rebuttal, improved the manuscript and **engaged in corresponding discussions with the reviewers including Reviewer xwPR, Reviewer aMu3 and Reviewer WbzD. They indicated their intention to raise their scores:** Reviewer WbzD had updated the score from 2 to 8 three days before scores were locked while the other two reviewers expressed their intention to increase their scores after reading the rebuttal, but the scores had already been locked. **Before the scores were reverted, the ratings were 10, 6, 6, 8.  And we strongly believe that had the scores not been locked, our final ratings would have reached 10, 8, 8, 8.** Below is a concise summary of the discussion:
1. Reviewer WbzD (score 2 -> 8) recognized the novelty of our core idea and the encouraging empirical results, then suggested addressing the lack of theoretical justification and providing comprehensive experiments to improve the current version of the paper. In response to these valuable comments:
- We **provided an extended and detailed discussion** of idempotence, explaining from a theoretical perspective how enforcing idempotence in continual learning improves calibration and mitigates catastrophic forgetting.
- We **conducted the suggested evaluation** by extending ECE metrics on Tiny-ImageNet and comparing post-hoc calibration methods in the continual learning setting, adding comparisons with regularization-based methods, and reporting results across different backbones. These results demonstrate the robustness of our approach.
- We **conducted the suggested comprehensive hyperparameter ablations**, further highlighting its stability and reproducibility.
- We incorporated the discussed experiments and **improved the manuscript** accordingly based on the above feedback.

  Consequently, reviewer WbzD provided positive response including address concerns and stating that these changes make **“the paper in a much better shape than before and can be a valuable contribution,”** and decided to **raise the score from 2 to 8** on November 25, **three days before the score was locked.**

2.  Reviewers xwPR (score 6 -> 8) and aMu3 (score 6 -> 8) initially assessed the paper as borderline accept and offered several insightful suggestions. In response, we **evaluated our method in additional settings** such as the online continual learning setting, **used additional datasets** containing out-of-distribution (OOD) samples to assess robustness, and **provided concrete examples** to illustrate how our method achieves error correction. **Both reviewers indicated an intention to raise their scores,  noting that their concerns had been addressed. However, after November 27, the scores had already been locked.**


Finally, in the interest of fairness, **we kindly ask that you consider the aforementioned review conditions while making the final decision.** We hope that our contributions,  a novel framework for continual learning based on idempotent property, which is a simple and robust approach that can be easily integrated into other state-of-the-art methods, receive careful consideration. In light of the above, we believe that our paper is at Oral level.

Thank you for your understanding!

---

### Meta-Review · Area_Chair_xT4U · 2026-01-02

**Summary:**

This paper received initial scores of 10, 6, 6 and 2.

The 10 was given by Reviewer f4Fu. Their motivation for this high rating is however rather short: "The paper is well written and organized. Adaptation of the idempotence loss function to the continual learning setting is creative and nuanced. The experimental results are promising." The confidence of this reviewer is also not so high (level 3) and the lowest among the four reviewers.

The 2 was given by Reviewer WbzD. The reviewer indicated that although they think the idea of the paper is interesting and the empirical results are encouraging, the paper still "suffers from significant theoretical and experimental gaps". The authors provided an extensive rebuttal to the review of Reviewer WbzD. After reading this rebuttal, as well as the reviews of the other reviewers, Reviewer WbzD indicated that their concerns had been addressed and that they had decided to raise their score. They indicated that they believe that the paper "can be a valuable contribution to the research community". Based on this, I expect that Reviewer WbzD would have raised their rating from 2 to 6, or perhaps to 8. (Note that in their rebuttal summary for the AC, the authors indicate that this reviewer had raised their score from 2 to 8, but it seems to me that I am not allowed to take this information into account.)

Reviewers xwPR and aMu3 both initially gave a rating of 6. Both reviewers raised the concern that the paper provided limited theoretical insight for why idempotence is beneficial for continual learning and uncertainty calibration, and they both further raised some concerns related to limited empirical evaluation. Following the author rebuttal, both reviewers (xwPR and aMu3) posted brief messages that their concerns had been addressed and that they intended to raise their scores. However, it seems important to note that both these messages seem to have been posted after it was announced the scores would be frozen (they were posted on 28 nov), so it is unclear to me whether I am allowed to take these messages into account or not.
If I am allowed to take these last messages into account, I expect both reviewers (xwPR and aMu3)  would have raised their rating from 6 to 8.
If I am not allowed to take these last messages into account, I expect both reviewers (xwPR and aMu3) would have maintained their rating of 6. The reason I would expect so is that I myself am not so convinced about the authors' rebuttal regarding the theoretical insight / motivation for why their proposed method works well (below, when I talk about my own reading of the paper, there is some motivation for this expectation).

Depending on what information I should take into account, the expected final ratings are therefore either 10, 6, 6, 6 or they are 10, 8, 8, 8. In both cases, it is good to mention that the 10 rating is not accompanied by a strong motivation.

In either case, there seems to be agreement among the reviewers that this paper should be accepted. Moreover, depending on what information should be taken into account, there is either some or quite strong reason to consider this paper for an oral presentation, or perhaps even for an award.

Based on my own reading of the paper, I would recommend it for acceptance, but not for an oral presentation or an award.
I agree with the reviewers that the paper's proposal (i.e., to use idempotence to create a new method for continual learning) is creative, and that the empirical results indicate that the method has merit. I therefore support acceptance of this paper. However, I share the concern of most reviewers that the paper provides rather limited theoretical insight for why the method is suitable for continual learning and uncertainty calibration, and I also think that the provided empirical evaluation provides rather limited evidence regarding a potential benefit for this method in practical problems. Unlike the reviewers, I do not think the author rebuttal satisfactorily addressed these concerns. I therefore only recommend acceptance of this paper as a poster, not as an oral.

**Reviewer Concerns:**

Please see above. This is integrated in my answer to the first question.

**Reviewer Scores:**

Please see above. This is integrated in my answer to the first question.

---

### Decision · Program_Chairs · 2026-01-26

Accept (Poster)